# Evaluating Language Models in Longer Conversational Contexts

## Abstract

Evaluating long-form conversations between humans and large language models (LLMs) poses significant challenges in the field of natural language processing. Traditional evaluation metrics and benchmarks have largely focused on shorter language interactions and often fail to capture the nuances inherent in extended dialogues. To address this, we introduce UPHELD, a publicly available dataset featuring human-annotated long-form dialogues. This dataset not only facilitates robust benchmarking but also serves as a foundation for further research into conversation evaluation methodologies. Using our dataset, we systematically analyze the correlation between current LLM evaluation metrics and human judgment within long-form conversation scenarios. Our findings reveal that conventional metrics lack the sensitivity necessary to assess the complex and often subjective nature of prolonged interactions. We use our dataset to develop an improved evaluation metric that demonstrates a significantly higher correlation with human assessments. The work highlights the need for advanced metric designs and outlines a clear pathway to refine the evaluation of LLM long-form conversations.

## 1 Introduction

Large Language Models (LLMs) have demonstrated remarkable capabilities in generating high-quality text, particularly in tasks requiring focused question answering. However, a critical and increasingly relevant application lies in open-ended conversational settings, where the goal is not to retrieve a specific answer but to engage in a natural, extended dialogue. This includes applications such as companionship and exploratory discussion, as well as complex business interactions such as customer service; these applications all require a shift from retrieval to genuine interaction.

Evaluating LLMs in these longer-form, open-domain conversations presents a significant challenge. Existing evaluation metrics (Rein et al., 2023; Hendrycks et al., 2021), largely inherited from tasks like question answering and machine translation, often focus on factual correctness, precision, and recall against predefined targets. These metrics prove inadequate when assessing the more nuanced qualities of a successful conversation, such as coherence, engagingness, personality consistency, and the ability to maintain context over multiple turns. Simply put, current metrics struggle to capture what makes a conversation feel natural and satisfying: the very qualities that define a positive human-LLM interaction and especially in a longer dialogue setting.

This inadequacy stems from the differences between conversations in the two length regimes. Unlike question answering, where success is defined by matching a ground truth, a 'good' conversation is subjective and multifaceted. It requires the LLM to demonstrate not just linguistic competence, but also social intelligence and an understanding of conversational dynamics. Existing metrics fail to model dialogue dynamics, human subjectivity, or align with goals like depth or reasonableness.

To address this gap, we reformulate the problem of evaluating long conversations as a *consistency measure* between the $k$th turn of a conversation with respect to the $1, 2, \ldots, k-1$th turns within an $n \geq k$ turn conversation. Rather than traditional measures of information accuracy, this consistency lens focuses our attention onto measures like long-term coherence and reasonableness, which are precisely the under-represented measures that are instrumental in the longer conversational domain.

UPHELD (Utility & Planning Human Evaluated Long Dialogues) contains over 10,000 high-quality human annotations across hundreds of novel long-length conversations focused on useful task

completions (e.g. math tutoring, trip planning, etc.). Each assistant turn in these conversations is labeled for coherence in content and style, as well as general reasonableness for a variety of different popular models responses. We will demonstrate that standard metrics perform poorly on the UPHELD dataset, and that both fine-tuning on UPHELD conversations and taking simple ensembles of standard metrics leads to large improvements. These results show that not only is UPHELD revealing a crucial evaluation modality (i.e. longer conversations) that challenges the standard methods, but also that it provides meaningful vectors by which to improve these this important modality. Our primary contributions are:

1) We present a dataset of human-evaluated long-form, non-topical conversations. UPHELD introduces difficult benchmarks such as content and style consistency that allow us to assess any proposed evaluation metric for its ability to align with subjective human opinions of natural dialogue interactions.

2) We apply UPHELD to show that common LLM evaluation metrics perform poorly within the long dialogue regime.

3) We demonstrate that ensembled machine-learned metrics trained on UPHELD exhibit stronger correlation with human judgments than most commonly used LLM automatic evaluation metrics. This result shows that our dataset enables meaningful exploration of language model improvements in our proposed setting (long conversation).

4) We show that fine-tuning a model for long-form conversations improves performance to surpass state-of-the-art results in this context.

## 2 RELATED WORK

Evaluating the performance of large language models (LLMs) is a critical area of research. Current benchmarks, such as those on the Open LLM leaderboard (now archived) (Myrzakhan et al., 2024), often focus on task-based evaluations using input-output metrics. These metrics include task-based generation evaluations like IFEval (Zhou et al., 2023) which evaluates the ability of LLMs to follow instructions and BBH (Big Bench Hard (Srivastava et al., 2022)), which comprises 23 groups of tasks such as *word sorting*, *casual judgment*, *navigate*. This and similar task-with-answer evaluations utilize metrics such as Exact Match (EM), Precision/Recall/F1, and sub-string based accuracy. Specific examples include MATH for mathematical problems (Hendrycks et al., 2021), and GPQA for multi-choice question answering (Rein et al., 2023).

Question answering (QA) benchmarks are also prevalent. MUSR (Sprague et al., 2023) is a narrative-based QA evaluation where the input is a paragraph and the output is an answer with an evaluation score. MMLU-PRO (Wang et al., 2024) combines task-based evaluation with QA and chain-of-thought prompting. Recently, using an LLM as a judge for evaluation is another approach which enables evaluating without having a ground truth or a reference answer (Clark et al., 2023) (Zheng et al., 2023), (Duan et al., 2023), (Adlakha et al., 2024). While the aforementioned benchmarks are prominent, several related areas of work are not typically included in leaderboards focused on single-turn evaluations, particularly those concerning longer conversations. These include approaches similar to Chatbot Arena (Zheng et al., 2023), MultiHop QA (Schnitzler et al., 2024) which involves answering questions over multiple documents or turns, and MT-eval which focuses on evaluating conversations directly and multi-dimensionally (Kwan et al., 2024). BotChat (Duan et al., 2023) is also noted as a method for conversation generation evaluation.

The above works often rely on input-output pairs, exact matches, or scores tied to specific correct answers, which are well-suited for evaluating ability to perform discrete tasks or extract factual information. However, they may not be appropriate for evaluating long, non-topical conversations because such interactions involve sustained coherence, context management over many turns, engaging dialogue, and the ability to handle subjective or open-ended discussions that do not have a clear and concise "correct" answer. Current long-form datasets focusing on non-factual conversations, including MuTal (Cui et al., 2020), Topical-chat (Gopalakrishnan et al., 2019), LLM-arena (Zheng et al., 2023), and DailyDialogue (Multi-turn) (Li et al., 2017), often lack authentic human-to-human interaction or are constrained to predefined topics. Recognizing these limitations, and to construct a robust benchmark, we identified the need for a dataset characterized by comprehensive curation,

expert authorship, and human annotation. This paper introduces such a dataset, which we have meticulously collected and organized, with details provided in the ensuing sections.

# 3 DATASETS

## 3.1 UPHELD DATASET

We first collected conversational data by tasking a panel of professional writers hired from the Upwork freelancer marketplace to create open-ended, naturalistic dialogues across a diverse range of topics like customer service and education. The writers were explicitly instructed to create longer dialogues characterized by conversational complexity and exploratory interaction, rather than producing conversations with predetermined or narrow outcomes. For example, the writers developed dialogues exploring scenarios such as mobile phone selection, travel planning, or problem-solving discussions that were goal-oriented and where multiple nuanced exchanges would be appropriate. Writers were also instructed to avoid overly scripted or linear communication patterns. This approach ensures that the dataset captures the diverse and intricate nonlinear nature of human conversation. We also took steps to alleviate any potential writer bias (see details in Appendix B). The UPHELD dataset is publicly available under a permissive CC-BY-4.0 license and an anonymized copy is provided in the supplemental materials of this submission, with the public download URL temporarily withheld to preserve anonymity during the review process.

### 3.1.1 INPUT DATA

Given our initial set of rich natural language conversations, various LLM models were then used to output candidate completions at every level of every conversation. Specifically, models were presented with conversation history up to a specific point, with the next human-written turn withheld. Models then generated a predicted next turn. The following models were used to generate the dataset: a) **GPT-3.5** (OpenAI, 2023); b) **GPT-4o** (OpenAI, 2024); c) **Llama-3.1-70b** (Grattafiori et al., 2024); d) **custom model** (a fine-tuned version of Llama-3.1: see Section 5.1). In addition to these, we also added two baseline candidate completions: a) a version of the ground truth (human) output rephrased using GPT-4o and b) a random completion. These two additional benchmarks offer important opportunities to calibrate our data, and as expected we found that the rephrased dataset received the highest marks while the random dataset received the lowest marks when evaluated by a human labeler. These checks provide us with additional confidence that humans are fair judges of our evaluation axes – style, content, and reasonableness.

At every possible conversational turn, annotators were then given (1) the chat history up to that point, (2) the ground truth human completion (Option A), and (3) one of the LLM-generated candidate completions (Option B). The provenance of options A and B were not disclosed to the annotators. Their task was to compare these continuations on content consistency, style consistency, and general reasonableness. The tasks, metric scales and instructions were fine tuned through (paid) two pilot user studies. Example data-points within UPHELD are given in Table 1, and full instructions provided to the annotators can be found in Appendix A.

### 3.1.2 ANNOTATION DIMENSIONS

Across all candidate models, each model-generated turn was evaluated by five independent human annotators using the following scoring criteria (see full instructions in Appendix A).

- Content Equivalence: A 5-point Likert scale (1-5) assessing semantic similarity between generated and reference turns.

- Style Equivalence: A 3-point Likert scale (1-3) measuring linguistic and stylistic correspondence.

- Utterance Reasonableness: A binary scale (1: unreasonable, 5: reasonable) evaluating the contextual appropriateness and coherence of the generated response,

| chat_history | *user*: I'm shopping for a new car and can't decide what to get.
*assistant*: I can help with that. What size car are you looking for? | *user*: I'm really frustrated. I've been trying to get this printer to work all morning and it just won't print anything!
*assistant*: I understand that must be really annoying. Let's work together to fix this. Can you tell me if you see any error messages on your printer or computer?
*user*: No error messages. It just sits there doing nothing whenever I try to print!
*assistant*: Alright, let's try a few steps to see if we can get it responding. First, could you check whether the printer is set as the default printer on your computer?
*user*: How do I check that? |
|---|---|---|
| **Option A** | The Ford F-150 is one of the top selling trucks in the country. Have you taken one of those for a test drive? | If you're using a Windows PC, you can go to 'Devices and Printers' in your control panel. For Mac, you can check under 'System Preferences' and then 'Printers & Scanners'. |
| **Option B** | Trucks are great for hauling and towing. Here are a few questions to help narrow down your options:
1. What will you be using the truck for most often? (e.g., work, off-roading, daily driving, towing a trailer)
2. How many passengers will the truck need to seat?
3. Do you have a preferred fuel type (gasoline, diesel, hybrid)?
4. What's your budget for the truck?
5. Do you have a specific brand or model in mind?
6. Do you need any specific features, such as 4-wheel drive, a certain bed size, or advanced safety features | It's easy. On a Windows computer, you can go to the Control Panel, then Devices and Printers (or Printers and Devices in some versions). Look for your printer's name and see if it has a check-mark next to it that says "Default printer". If it doesn't, right-click on the printer and select "Set as default printer".
On a Mac, you can go to System Preferences, then Printers & Scanners. Look for your printer's name and make sure it's selected as the default printer.
If you're still having trouble, we can try some other steps. Have you tried restarting the printer |
| **score_task_1_content** | 1: Strongly Disagree – no content overlap | 3: Neutral - some overlap of content |
| **score_task_2_style** | 1: Different styles | 1: Different styles |
| **score_task_3_reasonableness** | 1: Not a reasonable continuation | 5: A reasonable continuation |

Table 1: Input conversations and associated annotator ratings from a single annotator (all scores are out of 5). More examples can be found in Appendix K.

### 3.1.3 DATA STATISTICS

In total we collected complete evaluation labels for 400 turns stemming from 53 human written conversations collected for this study and additional seven conversations from existing datasets (Cui et al., 2020) to serve as control points. The dataset was curated to include approximately 90% human-written conversations on a wide range of topics and situations, supplemented with approximately 10% from available datasets to introduce challenging cases.

Each predicted turn was evaluated by five human annotators, and each annotator judge labeled between 1,220 and 1,230 conversations. We overall generated 12,291 sets of labels, or 36,873 labels. The ground truth conversations consist of 5.2 turn pairs (user-assistant) or 10.4 dialogue turns on

average. The average length of the conversation history annotators analyzed was 560 characters and the length of the judged turns was on average 245 characters.

### 3.1.4 VERIFICATION DATASETS

To validate our experiments, we constructed an independent verification datasets by augmenting LLM-Arena (Zheng et al., 2023) and Topical-Chat (Gopalakrishnan et al., 2019). The overall procedure consisted of three steps: (i) deriving a single "ground truth answer" from each data point of each existing dataset (see below), (ii) generating an alternative continuation with GPT-4o, and (iii) collecting human judgments following the UPHELD annotation protocol. We include all additional verification labels within our dataset for reproducibility.

Derivation of Ground Truth:

   a. LLM-Arena:
      – Data Point Description: each dialogue is accompanied by two model-generated continuations plus a human preference label.
      – Ground Truth Extraction: for every conversation we enumerated all candidate pairs, tallied human preferences, and chose the majority-preferred continuation as the reference (provided a clear winner existed).
   b. Topical-Chat:
      – Data Point Description: human-to-human dialogues grounded in specific topic selections.
      – Ground Truth Extraction: to mitigate cold-start artifacts, we extracted segments spanning turns 5–7. We then treated turns 1–5 (or 1–7) as the model input and selected the next human turn (turn 6 or 8) as the ground truth continuation.

After standard quality control (i.e. filtering for missing data and badly formatted inputs), we obtained 12,305 pairwise preference judgments. We note that both these verification datasets, although useful for verification, are still relatively deficient in freeform human-to-human interaction and focus on a limited set of pre-defined topics. As such they should be treated as verification datasets only and not as valid replacements for UPHELD.

## 4 METRICS

In total, 12 candidate metrics were assessed for their ability to evaluate longer conversations via correlation to UPHELD labels. These metrics were grouped into 3 distinct groups: 1) token-based – metrics quantifying similarity based on exact overlap of tokens 2) semantic-based – metrics quantifying similarity based on semantic overlap (e.g. embedding models); and 3) LLM-based – metrics employing some form of the LLM-as-a-judge paradigm.

Recall-Oriented Understudy for Gisting Evaluation (ROUGE (Lin, 2004)) is a set of standard language metrics that compares an automatically produced summary or translation against a set of reference summaries or translations. Specifically, ROUGE-N measures the overlap of n-grams between the system-generated text and the reference text. ROUGE-L measures the longest common subsequence, which accounts for sentence-level structure similarity.

We also explored cosine similarity between message embeddings as a measure of semantic similarity between the generated text and reference text. This approach is rooted in the work by Reimers & Gurevych (2019) on Sentence-BERT embeddings, which have shown effectiveness in capturing semantic similarities in text data. BERTScore leverages the pre-trained contextual embeddings from BERT to evaluate text generation by matching words in candidate and reference sentences. It computes precision, recall, and F1 score, providing a more nuanced evaluation than traditional n-gram based metrics.Zhang et al. (2020) introduced BERTScore as a robust metric for evaluating generated text.

We also tested LLM-as-a-judge metrics (Clark et al., 2023) against UPHELD. This approach involves using a separate LLM to score the outputs based on various criteria, such as coherence, relevance, and overall quality. We used both binary (yes/no) and Likert scale (1-5) judgments, with and without explanations. Prompts for judges can be found in Appendix F.

## 4.1 REFERENCE-FREE VS HUMAN-GROUNDED METRICS

Fundamentally, UPHELD uses a ground truth reference to generate our human labels. In contrast, reference-free evaluation of LLMs Liu et al. (2023) rely on human preference, and LLM outputs can reliably reproduce these preferences, indicating their performance is consistent with human judgments (Zheng et al., 2023). However, reference-free preference datasets also incur significant limitations, such as poor performance at judging long task-oriented dialogues and weakened judgment reliability on out-of-distribution tasks (Saito & Sato, 2025). UPHELD is designed specifically to tackle these limitations by adopting a reference-full approach.

However, the reference-full approach raises a question around dialogue multiplicity: a single input might incur multiple valid outputs, so how are we sure our ground truths are well defined? UPHELD's design reduces susceptibility to this issue in two ways: (1) two of our key label categories (style and reasonableness) are well-defined even with dialogue multiplicity, and (2) UPHELD dialogues primarily revolve around task-oriented settings, which means content accuracy is a well-defined metric. For example, while opinion-oriented conversations (e.g. *Who makes the best Caesar salad?*) are susceptible to dialogue multiplicity, our task-oriented dialogues (e.g. *How to make a Caesar salad?*) are not. To quantify this effect, we ran an experiment detailed in Appendix C and showed that UPHELD dialogues admit significantly higher output consistency (93%) compared to freeform dialogues (74%), which shows that our reference-full approach still allows us to collect meaningful labels on ground-truth content overlap.

## 5 EXPERIMENTS

To demonstrate the adds value to long-form conversational evaluation, we present a series of experiments. We show: a) that the base conversations within UPHELD can be used to greatly increase conversational fidelity within an LLM; b) that naive evaluation metrics fail in the long conversations, and propose a simple ensemble metric which greatly outperforms the baseline metrics. These ensemble metrics also perform well on our validation datasets (Topical and LLM Arena), showing that a method developed with UPHELD is transferrable to other contexts. We also include discussion on user disagreement within UPHELD.

## 5.1 UPHELD AS A FINE-TUNING DATASET

An important way to validate the quality of data within the UPHELD dataset is to assess how the UPHELD scores differ between the *custom* model, which was fine-tuned on the base data, and the other baseline models. To do so, we directly plot the mean content, style, and reasonableness scores within the UPHELD dataset for the different models in Figure 1. Base models (GPT-3.5, GPT-4o, and Llama3.1-70b-base) exhibit significantly lower performance in both content accuracy and style accuracy, with mean scores below 0.5. This indicates a tendency for these models to deviate from the intended conversational style and introduce content inconsistencies. Specifically, the base models demonstrate a substantial deficit in maintaining the stylistic integrity and topical coherence of the conversation, suggesting a propensity to diverge from the trajectory established by human writers.

In contrast, the custom model, fine-tuned for extended dialogue on a held-out set, shows a marked improvement and achieves approximately a 40% increase in both content and style accuracy compared to the base models. It does this while retaining the core functionality of its model base (Llama-3.1-70b), which we validated at test time and also at training time where we observed minimal overfitting within loss curves. Reasonableness is fairly flat across all models, which is not surprising as LLMs tend to output reasonable results regardless of style or content consistency. These results highlight that the UPHELD dataset's conversations are both learneable and encode behavior that are not well-exposed within these models' pre-training datasets.

## 5.2 EVALUATION METRIC PERFORMANCE ON UPHELD

Given that UPHELD is designed to help develop novel evaluation metrics for long conversations, it is instructive to see how this development works in practice. We start this section with an analysis on how traditional metrics perform poorly on UPHELD, and how simple modifications to the traditional

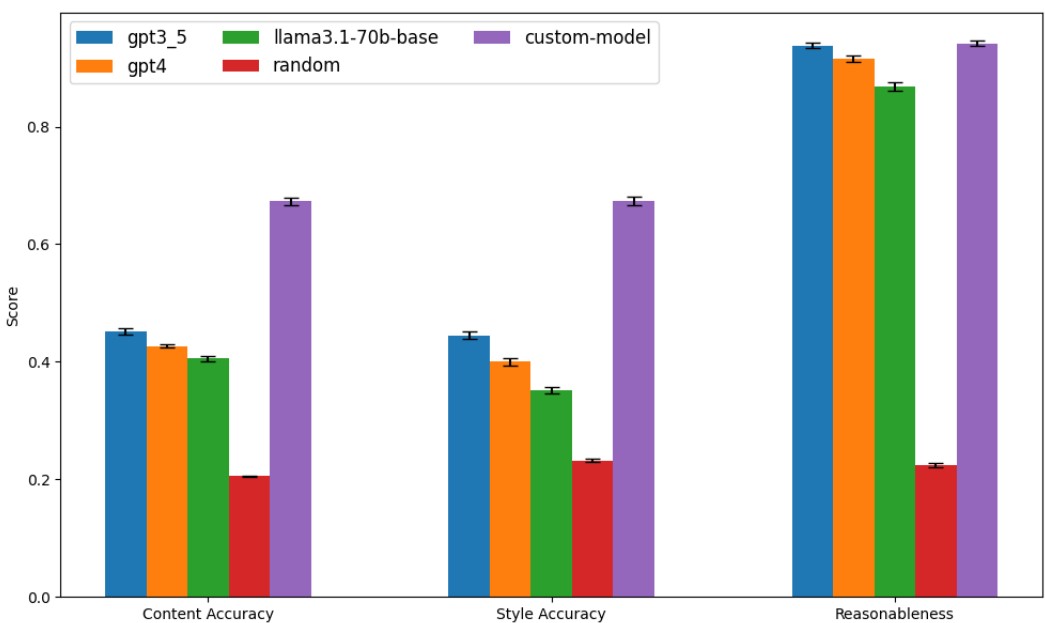

Figure 1: Aggregate human labeler scores as a share of the perfect score (see Appendix D.1) for each model on the UPHELD dataset within each label category. GPT-4 rephrase performs the best because it is a mild perturbation of the real ground truth. Of the other models, the custom model, which was fine-tuned on a held-out set of training data within the UPHELD dataset, performs significantly better than all other baseline models, demonstrating that the information to be learned within UPHELD conversations is significant.

metrics provide a significant boost in performance. All results in this section are standard correlation coefficients presented as 5-fold cross validation results on a 20% held-out set.

### 5.2.1 TRADITIONAL METRICS

The main results for traditional metrics (as defined in Section 4) are shown in the first three rows of Tables 2 and 3. The results reveal a weak to moderate correlation between metrics and human ratings. This suggests that no single metric captures the nuances of human assessment well within the UPHELD dataset. All definitions for metrics can be found in the Appendix D.2. These results reveal a weak to moderate results for these metrics. This suggests that no single traditional metric captures the nuances of human assessment well within the UPHELD dataset. Interestingly, the semantic metrics (like bert_score_F1) demonstrated the highest correlation with human judgments for both content and style aspects. The token-based metrics showed stronger correlations than LLM-as-a-judge as well. These findings suggest that LLM-as-a-judge, despite being increasingly explored in the literature (e.g., (Ma & Graham, 2019) (Zhang et al., 2020)), is a weak evaluator of longer conversations.

In spite of the observed strength of the semantic metrics, the results for traditional metrics show weak to moderate performance on UPHELD. This implies that relying solely on any single traditional metric inadequately captures the complexities of content and style quality in longer conversations, and we demonstrate that ensemble metrics can bridge this gap.

### 5.2.2 ENSEMBLE METRICS

We hypothesize that individual metrics attend to distinct facets of text quality, and so a learned ensemble will perform better on the UPHELD dataset. We train linear regression, SVM, and random forest models to predict human scores using the individual automatic metric scores as input features. Using this we derive a new, hybrid metrics and assess their correlation with human judgments.

The last 3 rows of Tables 2 and 3 show that the learned ensemble metrics exhibit higher correlations with human scores compared to any single metric in isolation. Notably, the random forest model

| Metric | UPHELD | LLM Arena | Topical Chat |
|---|---|---|---|
| **Semantic Metrics** | | | |
| message_embedding_cos_sim | $0.58\pm_{0.003}$ | $0.41\pm_{0.004}$ | $0.40\pm_{0.008}$ |
| bert_score_precision | $0.36\pm_{0.005}$ | $0.39\pm_{0.004}$ | $0.32\pm_{0.008}$ |
| bert_score_recall | $0.40\pm_{0.005}$ | $0.42\pm_{0.004}$ | $0.40\pm_{0.008}$ |
| bert_score_F1 | $0.40\pm_{0.005}$ | $0.45\pm_{0.003}$ | $0.42\pm_{0.008}$ |
| **Token-based Metrics** | | | |
| rouge1 | $0.34\pm_{0.005}$ | $0.49\pm_{0.003}$ | $0.33\pm_{0.009}$ |
| rouge2 | $0.24\pm_{0.005}$ | $0.43\pm_{0.003}$ | $0.24\pm_{0.009}$ |
| rougeL | $0.32\pm_{0.005}$ | $0.41\pm_{0.004}$ | $0.34\pm_{0.010}$ |
| rougeLsum | $0.31\pm_{0.005}$ | $0.50\pm_{0.003}$ | $0.34\pm_{0.009}$ |
| **LLM-as-a-judge metrics** | | | |
| llm_judge_yes_no | $0.40\pm_{0.012}$ | $0.32\pm_{0.015}$ | $0.57\pm_{0.016}$ |
| llm_judge_yes_no_explain | $0.23\pm_{0.018}$ | $0.32\pm_{0.014}$ | $0.29\pm_{0.013}$ |
| llm_judge_likert_1_5 | $0.26\pm_{0.008}$ | $0.41\pm_{0.011}$ | $0.39\pm_{0.014}$ |
| llm_judge_likert_1_5_explain | $0.28\pm_{0.009}$ | $0.41\pm_{0.013}$ | $0.34\pm_{0.013}$ |
| **Ensembled ML Metrics (Ours)** | | | |
| Linear Regression | $0.59\pm_{0.004}$ | $\mathbf{0.50}\pm_{0.003}$ | $\mathbf{0.58}\pm_{0.006}$ |
| SVM | $0.55\pm_{0.005}$ | $0.49\pm_{0.003}$ | $0.54\pm_{0.008}$ |
| Random Forest | $\mathbf{0.63}\pm_{0.004}$ | $0.23\pm_{0.004}$ | $0.32\pm_{0.009}$ |

Table 2: Results on the UPHELD datasets for various candidate evaluation metrics on the first evaluation task (content accuracy). Ensemble metrics clearly perform better for both the UPHELD dataset and both verification datasets.

.

| Metric | UPHELD | LLM Arena | Topical Chat |
|---|---|---|---|
| **Semantic Metrics** | | | |
| message_embedding_cos_sim | $0.41\pm_{<0.001}$ | $0.42\pm_{0.004}$ | $0.29\pm_{0.009}$ |
| bert_score_precision | $0.37\pm_{0.005}$ | $0.39\pm_{0.004}$ | $0.36\pm_{0.008}$ |
| bert_score_recall | $0.36\pm_{0.005}$ | $0.50\pm_{0.004}$ | $0.27\pm_{0.009}$ |
| bert_score_F1 | $0.41\pm_{0.005}$ | $0.49\pm_{0.003}$ | $0.35\pm_{0.008}$ |
| **Token-based Metrics** | | | |
| rouge1 | $0.32\pm_{0.005}$ | $0.51\pm_{0.003}$ | $0.31\pm_{0.010}$ |
| rouge2 | $0.20\pm_{0.005}$ | $0.43\pm_{0.004}$ | $0.16\pm_{0.010}$ |
| rougeL | $0.29\pm_{0.005}$ | $0.48\pm_{0.004}$ | $0.30\pm_{0.009}$ |
| rougeLsum | $0.29\pm_{0.005}$ | $0.48\pm_{0.003}$ | $0.30\pm_{0.009}$ |
| **LLM-as-a-judge metrics** | | | |
| llm_judge_yes_no | $0.35\pm_{0.012}$ | $0.08\pm_{0.010}$ | $0.36\pm_{0.016}$ |
| llm_judge_yes_no_explain | $0.15\pm_{0.015}$ | $0.08\pm_{0.010}$ | $0.24\pm_{0.013}$ |
| llm_judge_likert_1_5 | $0.24\pm_{0.010}$ | $0.07\pm_{0.009}$ | $0.22\pm_{0.014}$ |
| llm_judge_likert_1_5_explain | $0.20\pm_{0.012}$ | $0.05\pm_{0.008}$ | $0.20\pm_{0.013}$ |
| **Ensembled ML Metrics (Ours)** | | | |
| Linear Regression | $0.50\pm_{0.004}$ | $0.28\pm_{0.004}$ | $0.32\pm_{0.008}$ |
| SVM | $0.49\pm_{0.005}$ | $\mathbf{0.54}\pm_{0.003}$ | $\mathbf{0.55}\pm_{0.009}$ |
| Random Forest | $\mathbf{0.61}\pm_{0.004}$ | $0.04\pm_{0.004}$ | $0.22\pm_{0.009}$ |

Table 3: Results on the UPHELD datasets for various candidate evaluation metrics on the second evaluation task (style accuracy). Ensemble metrics clearly perform better for both the UPHELD dataset and both verification datasets.

yields substantial improvements of 30-40% relative to the best-performing individual metrics on UPHELD, but an SVM ensemble produces consistently higher correlations to human judges across all datasets. Note that the ensemble metrics were trained only on the UPHELD data and then applied to the LLM Arena and Topical Chat data. Although the transferability of the random forest model was

| models: | gpt3_5 | gpt4 | gpt4_rephrase | llama3.1-70b-base | random | custom-model |
|---|---|---|---|---|---|---|
| Kappa score | 0.30 | 0.33 | 0.46 | 0.38 | 0.97 | 0.35 |

Table 4: Kappa scores for different models. A Kappa score ranges from -1 to +1, with +1 indicating the maximum level of agreement among the annotators.

poor, the SVM and linear regression ensembles indicate that remarkably, trained metrics developed just on the UPHELD dataset have exceptional transferability to other out-of-domain datasets.

The success of the ensemble metrics likely stems from their ability to integrate diverse signals captured by the individual metrics, mirroring the multifaceted nature of human evaluation. These findings strongly suggest that within complex settings like long dialogue, learning to ensemble multiple automatic metrics offers a promising avenue for developing evaluation frameworks that more closely align with human judgments than relying on a single metric.

### 5.3 DISCUSSION AROUND ANNOTATOR AGREEMENT

Because each conversation turn was independently labeled by five human annotators, we also analyze model responses that induced stronger human-human agreements. We observed moderate levels of consensus as shown in Table 4, which tabulates Kappa scores (Cohen, 1960) between raters. As expected, human-human agreement is highest on the random and gpt4_rephrase baselines, while being consistent across all other models. Further analysis on labeler agreement can be found in Appendix G.

We observed that approximately 25% of data points had full agreement across all 5 human judges. Otherwise, we bin the level of agreement as follows: agreement across 2 out of 5 labels represent a "plurality," while agreement across 3 or 4 labels represent a "majority." We further quantify how well an LLM-as-a-judge evaluator agrees with this winning score relative to the agreement bin. Table 5 shows that as expected, LLM-as-a-judge performance is heavily correlated to agreement level amongst the human labelers. This result demonstrates that human-human disagreement is a valid measure of data difficulty, and this uncertainty signal present in UPHELD may be integral in further evaluation metric development.

| | LLM-as-a-judge | | | | |
|---|---|---|---|---|---|
| | human | likert | likert explain | binary | binary explain |
| plurality | 0.4733 | 0.3401 | 0.3696 | 0.7493 | 0.7196 |
| majority | 0.6932 | 0.3885 | 0.3976 | 0.8011 | 0.7674 |
| full agreement | 1.0000 | 0.6370 | 0.5603 | 0.9339 | 0.9195 |

Table 5: LLM-as-a-judge performance on the UPHELD dataset for different dataset splits separated by the level of human annotator agreement within each bucket.

## 6 CONCLUSION

In this work, we introduced UPHELD: a dataset designed to evaluate LLMs in long-form conversational settings. We collected tens of thousands high-quality human-annotated labels on crucial consistency metrics within long-form conversations, along with the high-quality conversations themselves. Analysis of existing evaluation metrics on UPHELD reveals that they do not effectively capture the nuances of human judgment for assessing conversational quality. We further demonstrated that simple-to-learn ensemble metrics result in substantially improved correlations with human evaluations. Taken in aggregate, our findings highlight the potential for developing robust evaluation frameworks that better align with human perceptions of effective conversation using UPHELD.

## 7 REPRODUCIBILITY STATEMENT

We provide the entire UPHELD dataset that was used for this work in order for any other researcher or practitioner to reproduce our work. Otherwise, all prompts used to collect our dataset are provided in Appendix F, and exact definitions of our baselines and models used are provided in Section 4 and 5. Explicit formulae to ensure baseline reproducibility are provided in Section D.2.

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

## A    ANNOTATION MATERIALS AND INSTRUCTIONS

Here we provide a complete description of the materials provided to data annotators along with associated instructions.

### A.1    MATERIALS

Users are given access to a spreadsheet file with three sheets – the first sheet contains the instructions, the second contains sample annotated examples and the third one contains a formatted table with 6 columns with the following labels:

- chat history – (the conversation up to one point)
- Option A – (one possible continuation to the conversation within chat history)
- Option B – (alternative possible continuation to the conversation within chat history)
- score_task_1_content – (a dropdown menu to select the content consistency score)
- score_task_2_style (a dropdown menu to select the style consistency score)
- score_task_3_reasonableness (a dropdown menu for choosing the reasonbleness score)

A screenshot of the interface annotators are given is shown in Figure 2.

### A.2    INSTRUCTIONS

This section contains the verbatim instructions provided to the annotators in the data annotation spreadsheet. We start with the overall instructions and then reprint the granular instructions for each of the three types of labels within UPHELD.

==BEGIN INSTRUCTIONS==

Given a chat history, you will be presented two options for how to continue the conversation: Option A and Option B. You will be asked to rate these options by answering a number of questions.

Task 1 (Content Equivalence): Do you agree that the general information presented in Option A roughly the same as the general information presented in Option B?

Task 2 (Stylistic Equivalence): Do you agree that the style of Option A the same as the style of Option B? Put another way, does it feel like Option A and Option B are being spoken by the same person?

Task 3 (Reasonableness of Option B): For Option B only, do you agree that Option B a reasonable way to continue the conversation, given the chat history?

Enter your score in the column corresponding to the task in the annotations tab/sheet (e.g. score_task_2_style).

Please read below for specific instructions and tips on each individual task.

#### A.2.1    TASK 1 – CONTENT EQUIVALENCE

Check the examples tab for some already annotated data and additional explanation (note that you are not expected to provide explanations of your scores.)" "Provide one of the following scores on a scale of 1-5 where a 1 reflects a strong DISAGREE and a 5 reflects a strong AGREE:

- 1: Strongly Disagree (that the content conveys equal information in both options)
- 2: Disagree
- 3: Neutral
- 4: Agree
- 5: Strongly Agree

Use the following criteria to help you determine if the two message options have equivalent content:

- Information conveyed by Option B contains all information that is conveyed by Option A.

- Using either Option A or Option B to continue the conversation would not change the flow of the conversation.

- You can replace Option A with Option B, or replace Option B with Option A, without appreciably changing the content of the conversation.

- Both Option A and Option B mean the same thing.

Tips:

- Keep the chat history in mind when considering the content equivalence of Option A and Option B.

- If one of the options seems incomplete or cut short, still try to evaluate the option as is.

- If Option B is wordier or contains more details than Option A, but it still contains all the information in Option A and is relevant given the chat history, lower the score to at most a 3.

- Do not lower the score if Option B contains AI self identification phrases such as (""As an AI agent..."", ""I am a trained model..) and similar. Focus on the other information within Option B.

- If Option B is not readable and/or contains non-coherent language give a score of 3.

- Lower the score if Option B contains more details that are (1) not an expansion of the information in Option A and (2) are not relevant to the messages in the chat history.

### A.2.2 TASK 2 - STYLE EQUIVALENCE

Provide one of the following scores on a scale of 1-5 where a 1 reflects a strong DISAGREE that styles are the same and a 5 reflects a strong AGREE that styles are the same:

- 1: Different styles (that the content conveys equal information in both options).
- 3: Somewhat same styles.
- 5: Same styles

Use the flowing instructions to help you determine if the two message options are stylistically equivalent

- After reading them out loud, both options A and B feel like they are written by the same person in the same mood.

- There is no noticeable change in sentiment or tone between the two options.

- Even if one of the message options is longer than the other, they can still be considered stylistically similar if the content is expressed in similar ways.

- If it sounds like option A and option B were written by different people, assign a low score.

- If you believe that both options are written by the same person in the same mood but the content of the two options are different, still assign a high score.

Tips:

- It may be useful to consider the context of chat history as a reference and seeing whether either option deviates from a natural continuation of the chat history, given the personality of the "assistant" in the chat history.

- For this task you're highly encouraged to read both options out loud as it may be helpful in forming the comparison.

- Consider differences in vocabulary, tone, and syntax when making your decision.

### A.2.3 TASK 3 – REASONABLENESS

Provide one of the following scores:

- 1: Not a reasonable continuation (to chat_history)
- 5: A reasonable continuation (to chat_history)

Guidelines:

- This task ONLY applies to Option B. The task it to determine whether Option B is a reasonable way to continue the conversation from the chat history.
- Ignore Option A in your judgment; Option B may be completely different from Option A but still score highly in this task as long as it is on topic.
- If Option B seems cut short assess the text up to the cutoff point.
- If Option B is not readable and/or sounds incoherent assign a score of

Tips:

- We encourage you to read the chat history out loud as well as the message in Option B directly afterwards. If it sounds like a natural conversation flow out loud the score is likely a high score.
- If you were the "user" in this scenario and received Option B as the next response, would you be generally happy with the state of the conversation? If the answer is yes, the score is likely 5. If not the score is likely 1.
- Do not lower your score if Option B contains any model self identification (e.g. As an AI model....) but is still a viable continuation of chat history.
- All of the following reasons are valid for assigning a low score of 1:
  * Option B is excessively wordy and/or provides too much information.
  * Option B is incoherent.Option B seems random and gets off topic.
  * Option B is excessively rude or aggressive.
  * Option B has an inappropriate tone or uses inappropriate language.
  * Option B does not add any additional helpful information to the conversation or prompt the user to provide additional relevant information.
- Check the examples tab for some already annotated data and additional explanation (note that you are not expected to provide explanations of your scores.)

==END INSTRUCTIONS==

## B    NOTES ON WRITER AND ANNOTATOR SELECTION AND BIAS MITIGATION

Our contracted writers were required to have a high job success rate on Upwork and all were first evaluated through a rigorous initial (paid) pilot phase where their written conversations were evaluated by a professional user experience team for diversity and faithfulness to our prompts. Writers were also selected from diverse professional backgrounds: we employed writers with backgrounds from novel writing to education to copywriting. Prompts were selected for diversity of tasks and diverse defined styles that had to adhere to a number of user personas and styles. All conversations were quality-checked by a separate set of experienced proofreaders to explicitly ensure style diversity and consistency. We also acknowledge that our current focus is primarily on English language conversations, but also plan to eventually incorporate multilingual UPHELD additions.

All conversations were further quality-checked by another professional writer to ensure situational and stylistic diversity. We were admittedly limited to English-speaking writers, which may introduce some clustering of labeler backgrounds. Because each conversation went through multiple rounds of checks from different professionals (including both user research professionals, other writing professionals, and machine learning professionals) who were explicitly instructed to check for diversity and to

eliminate bias, we hope that any effects of geographical/linguistic clustering are mitigated by our rigorous process.

All data labelers also participated in an initial (paid) pilot program that was carefully evaluated by internal user research professionals before being selected to write conversations at scale. The scenarios the writers built were evaluated by the same user research professionals to ensure they covered a wide variety of scenario types and user behavioral/personality patterns which were representative of what chat agents might encounter in a customer-facing context.

## C  UPHELD AND CONVERSATIONAL MULTIPLICITY

A valid concern may be that direct comparisons to a reference human answer may be inappropriate in settings when a particular prefix can lead to a multiplicity of valid responses. This effect may be prevalent especially when the prefix is asking for an opinion (e.g. "What is your favorite animal?"). We however observe that UPHELD dialogues avoid this potential pitfall as they are not strictly freeform, but are all targeted towards completion of a specific well-defined task. In this context, there is some notion of correct ground truth, and we specifically hired professionals who are experts at these tasks (see Appendix B). To put it simply, our task setting is analogous to the difference between *what kind of salads do you like?* (which has ambiguity and dialogue multiplicity) and *how do I make a Caesar salad?* (which is much more constrained and has a more well-defined ground truth). To quantify this, we generated 100 open-ended questions (GPT-4o) and then generated two possible completions with GPT-4o at moderately high temperature ($\tau = 1$) to those questions. We did the same with 100 UPHELD turn completions. We then asked GPT-4o to judge whether the two possible completions contain similar content. The results are as follows:

| Dataset | Semantic Consistency (%) |
|---|---|
| Open-Ended | 74 |
| UPHELD (first turn only) | 93 |
| UPHELD (random turn) | 93 |

Table 6: Semantic Consistency Performance across Different Datasets

UPHELD induces much higher semantic consistency in the output, which means that UPHELD dialogues admit much less conversational multiplicity than more freeform datasets. This result supports our hypothesis that targeted task-focused conversations like the ones in UPHELD admit well-defined "ground truth" references. Interestingly, UPHELD maintains high output consistency even when we only analyze the first turn, which is where we would expect more branching/multiplicity during a dialogue. Note that these results are likely an underestimate of the true consistency, since sampling multiple LLM outputs would induce additional randomness that likely would not exist within natural human dialogue.

## D  METRICS

### D.1  AGGREGATION METRIC

The aggregate scores in Figure 1 represent the total score of a given response divided by the maximum possible sum score. If 3 judges score a turn 5, 3, and 2 with a maximum score of 5, the aggregate score is $(5 + 3 + 2)/(5 + 5 + 5) = 10/15$. More formally, Let a response be scored by $J$ judges. Judge $j$ gives a score $s_j$ with a per-judge maximum $M_j$ (often all $M_j = M$).

$$\text{score} = \frac{\sum_{j=1}^{J} s_j}{\sum_{j=1}^{J} M_j}, \quad \text{where } 0 \leq \text{score} \leq 1$$

### D.2  EXPERIMENTAL METRIC

We include all metrics used within the experimental studies tabulated in Tables 2 and 3.

- **Message Embedding Cosine Similarity:**

$$\text{CosineSim}(\mathbf{u}, \mathbf{v}) = \frac{\mathbf{u} \cdot \mathbf{v}}{\|\mathbf{u}\|\|\mathbf{v}\|}$$

where $\mathbf{u}$ and $\mathbf{v}$ are the embedding vectors of the reference and generated messages.

- **BERTScore Precision:**

$$P = \frac{1}{|x|} \sum_{i=1}^{|x|} \max_j \text{sim}(x_i, y_j)$$

- **BERTScore Recall:**

$$R = \frac{1}{|y|} \sum_{j=1}^{|y|} \max_i \text{sim}(x_i, y_j)$$

- **BERTScore F1:**

$$F1 = \frac{2PR}{P + R}$$

where $x$ and $y$ are the sets of tokens from the candidate and reference texts respectively, and $\text{sim}(x_i, y_j)$ denotes cosine similarity between contextual embeddings of tokens $x_i$ and $y_j$.

- **ROUGE-1 (Unigram Overlap):**

$$\text{ROUGE-1} = \frac{\sum_{w \in \text{Ref}} \min(\text{Count}_{\text{gen}}(w), \text{Count}_{\text{ref}}(w))}{\sum_{w \in \text{Ref}} \text{Count}_{\text{ref}}(w)}$$

- **ROUGE-2 (Bigram Overlap):**

$$\text{ROUGE-2} = \frac{\sum_{b \in \text{Ref}} \min(\text{Count}_{\text{gen}}(b), \text{Count}_{\text{ref}}(b))}{\sum_{b \in \text{Ref}} \text{Count}_{\text{ref}}(b)}$$

- **ROUGE-L (Longest Common Subsequence - LCS):**

$$\text{ROUGE-L} = \frac{\text{LCS}(X, Y)}{\text{Length}(Y)}$$

where $X$ and $Y$ are sequences of tokens in the generated and reference texts respectively.

- **ROUGE-Lsum (LCS over multiple sentences):**

$$\text{ROUGE-Lsum} = \frac{\sum_i \text{LCS}(X_i, Y_i)}{\sum_i \text{Length}(Y_i)}$$

where $X_i$ and $Y_i$ are sentence-level pairs from the candidate and reference summaries.

To compute all metrics, we used the ground truth next turn (*Option A*) as a reference data point and the model-generated next turn (*Option B*) as a candidate data point. We use the **mixedbread-ai/mxbai-embed-large-v1**[1] embedding model for all metrics that required a calculated similarity score.

All prompts associated with the *llm-as-a-judge* metrics can be found in Appendix F.

## E  HOW DID GPT-4O UNDERPERFORM GPT-3.5?

In Figure 1, we revealed a somewhat surprising result: humans tended to prefer the output of GPT-3.5 over that of GPT-4o. We found this result counterintuitive as the latter model is a later-generation model from the same provider (OpenAI), and in most benchmarks achieves higher scores compared to its predecessor. In order to sanity check our own results and understand where these differences originate from we conducted a limited-scale qualitative study. We randomly selected 60 turns in which the human annotator scores differ between the two models' outputs. These were then additionally judged across two dimensions: (a) general phrasing quality (evaluators could select between *human* or *template* sounding, and (b) perceived differences between the GPT-x output and the reference ground-truth output (evaluators were free to enter any difference). We then aggregated the results of this exercise, extracted major difference axes as described in the (b) labels, and we display the results in Figure 3. The Figure suggests that *GPT-4o* tend to be more verbose and less engaging in a conversation than GPT-3.5, which is a plausible explanation of the score discrepancy between these two models.

---
[1]https://huggingface.co/mixedbread-ai/mxbai-embed-large-v1

# F LLM-AS-A-JUDGE PROMPTS

In this section we reproduce the prompts used by the *llm-as-a-judge* metrics for our experiments that generated Tables 2 and 3.

- **llm judge - yes/no:**

  ```
  Conversation: {chat_history}

  Continuation: {Option A}

  Prediction:  {Option B}

  For the above your job is to compare the continuation and
  reference response as
  being equivalent in regards to the conversation.
  Output "Yes" if you think the continuation contains the same
  information as
  reference, otherwise output "No".
  ```

- **llm judge - yes/no explain:**

  ```
  Conversation: {chat_history}

  Continuation: {Option A}

  Prediction: {Option B}

  For the above your job is to compare the continuation and
  reference response
  as being equivalent in regards to the conversation.
  Output "Yes" if you think the continuation seems natural and
  human generated,
  otherwise output "No".
  Also output the explanation of why you made the judgment.
  ```

- **llm judge - likert 1-5:**

  ```
  Conversation: {conversation}

  Reference: {reference}

  Prediction:  {prediction}

  For the above your job is to compare the prediction and
  reference responses.
  Score whether the prediction conveys the same information as
  the reference the on
  a likert scale of 1 to 5.
  1 means none of the reference information is conveyed by the
  prediction;
  and 5 means reference and prediction are semantically
  equivalent.
  Output only scores from 1 to 5 (integer)
  ```

- **llm judge - likert 1-5 explain:**

  ```
  Conversation: {chat_history}

  Reference: {Option A}

  Prediction:  {Option B}
  ```

```
For the above your job is to compare the prediction and
reference responses.
Score whether the prediction conveys the same response
as the reference the on a score of 1 to 5 and give a reason
as to why.
```

## G  ANNOTATOR AGREEMENTS LEVELS

In addition to the Kappa scores that we presented in Section 5.3, we assessed annotator agreement through categorical bins to further analyze our dataset statistics. We quantified agreement at three distinct levels:

- **perfect** — where all annotators assign the same score to the same data point.
- **majority** — where more than half of the annotators assign the same score to the same data point.
- **lead/plurality** — where there is a score assigned more frequently than others to the same data point.

The results, as depicted in Figure 4, indicate that a substantial dataset can be retained even when considering only those data points on which all five annotators agree. Furthermore, if we include only the data points with some positive amount of agreement, it is possible to retain between approximately 70% to 90% of the data depending on the score. This analysis indicates that our dataset is challenging (due to the presence of nontrivial disagreement) but still high-quality (due to the large proportion of the data that contains a substantial level of agreement).

Additionally, we can show that more agreement between humans leads to less difficult tasks for the metrics. This is clearly shown in Table 7 where we show how the correlation between human and automatic metrics increases as the human agreement levels.

| metric | type | all | plurality | $\Delta\%$ | majority | $\Delta\%$ | perfect | $\Delta\%$ |
|---|---|---|---|---|---|---|---|---|
| dt_score | content | 0.59 | 0.84 | +42.4 | 0.79 | -6.0 | 0.90 | +13.9 |
| | style | 0.52 | 0.74 | +42.3 | 0.76 | +2.7 | 0.88 | +15.8 |
| lin_reg_score | content | 0.47 | 0.71 | +51.1 | 0.73 | +2.8 | 0.77 | +5.5 |
| | style | 0.28 | 0.58 | +107.1 | 0.64 | +10.3 | 0.66 | +3.1 |
| svm_score | content | 0.35 | 0.56 | +60.0 | 0.69 | +23.2 | 0.89 | +29.0 |
| | style | 0.14 | 0.49 | +250.0 | 0.56 | +14.3 | 0.67 | +19.6 |
| llm_judge_likert_1_5 | content | 0.21 | 0.33 | +59.5 | 0.37 | +10.6 | 0.45 | +21.2 |
| | style | 0.12 | 0.33 | +159.3 | 0.41 | +24.4 | 0.46 | +12.5 |
| llm_judge_yes_no | content | 0.33 | 0.53 | +59.3 | 0.59 | +9.8 | 0.83 | +41.4 |
| | style | 0.16 | 0.41 | +145.3 | 0.50 | +22.6 | 0.54 | +6.6 |
| rougeL | content | 0.33 | 0.56 | +69.7 | 0.59 | +5.4 | 0.68 | +15.3 |
| | style | 0.24 | 0.49 | +104.2 | 0.57 | +16.3 | 0.65 | +14.0 |
| bert_score_F1 | content | 0.33 | 0.59 | +78.8 | 0.64 | +8.5 | 0.65 | +1.6 |
| | style | 0.25 | 0.55 | +120.0 | 0.63 | +14.5 | 0.66 | +4.8 |

Table 7: Comparison of content and style scores with relative increases between agreement levels. The $\Delta$ values show the relative improvement on the previous level of human agreement.

# H    ADDITIONAL LINEAR REGRESSION ANALYSIS

In Tables 2 and 3 we showed that an ensembled linear regression classifier readily wins against single metrics. Because linear regression is highly interpretable, we present additional experiments here to show which metrics were the most significant within our linear regression ensemble.

We see in Figure 6 and 8 the p-value significance of each metric included in the ensemble. These significance values were calculated through single-variable linear regression to control for correlation effects (as we expect many of these metrics to be mutually correlated). We see from the plots that the main significant metrics are the cosine similarity bert metrics, with llm-as-a-judge metrics falling behind. Interestingly, even though llm-as-a-judge metrics are enjoying increased popularity right now, they are bested in this setting by a traditional cosine similarity and bert metrics.

From the actual coefficient values as shown in Figures 5 and 7 for content consistency, we see that other than the reasonableness llm-as-a-judge metric (which performed poorly and we omitted from most analysis within this work), all metrics have strong positive correlations with the UPHELD labels.

# I    THE REASONABLENESS LABEL

In the main paper, we provided extensive analysis of the accuracy and style UPHELD label sets, but UPHELD also contains a third set of labels around reasonableness. For completeness, we include the same analysis for the reasonableness label here, in Table 8. We also provide the same linear regression analysis as in Section H for the reasonableness label in Figures 9 and 10.

| Metric | UPHELD | LLM Arena | Topical Chat |
|---|---|---|---|
| **Semantic Metrics** | | | |
| message_embedding_cos_sim | 0.01 | 0.28 | 0.01 |
| bert_score_precision | 0.17 | 0.15 | **0.13** |
| bert_score_recall | 0.07 | 0.15 | 0.04 |
| bert_score_F1 | 0.14 | 0.19 | 0.10 |
| **LLM-as-a-judge Metrics** | | | |
| llm_judge_yes_no | 0.13 | 0.16 | 0.09 |
| llm_judge_yes_no_explain | 0.06 | 0.16 | 0.00 |
| llm_judge_likert_1_5 | 0.03 | 0.26 | 0.00 |
| llm_judge_likert_1_5_explain | 0.00 | 0.26 | 0.00 |
| **Token-based Metrics** | | | |
| rouge1 | 0.12 | **0.31** | 0.06 |
| rouge2 | 0.11 | 0.24 | 0.04 |
| rougeL | 0.12 | 0.18 | 0.07 |
| rougeLsum | 0.11 | 0.28 | 0.06 |
| **Ensembled ML Metrics (Ours)** | | | |
| Linear Regression | 0.16 | -0.16 | 0.01 |
| SVM | 0.01 | 0.20 | 0.03 |
| Random Forest | **0.52** | 0.18 | 0.04 |

Table 8: Reasonableness results on the UPHELD dataset and verification datasets.

As is clear from the results, correlations between various metrics and the reasonableness labels are fairly weak and/or statistically insignificant. Even though our ensemble tree model still performs admirably in this setting, the labels themselves have a very lopsided distribution with most labels being in the positive class (see Figure 1).

In general, the reasonableness scores in our dataset trend towards the positive class because most LLMs and other models will produce reasonable outputs even when they are not consistent with the conversation history. As in Figure 1, one can see that all models (except for the random model baseline) produce reasonableness scores that are substantially greater than 80%.

Due to both of these effects (the lopsidedness of the data and the lack of statistical significance in the regression results), we generally consider the reasonableness score as a sanity check label and a good filter for data that is out of distribution. It is for this reason that we decided to not analyze the reasonableness labels at length within the main paper. However, the reasonableness scores are still informative and we look forward to followup work to analyze this signal as a potential uncertainty or out-of-distribution feature.

## J   MORE DATASET STATISTICS: SCORE DISTRIBUTIONS

We provide more granular breakdowns of score distributions within the UPHELD dataset within this section, in Figures 11 (a)-(c).

## K   DATA EXAMPLES

On the following pages we now provide a number of other examples from the UPHELD dataset.

—

## L   LLM USAGE DISCLOSURE

We used LLMs to check for spelling errors but did not use LLMs to directly generate any text that appear in this paper. We also used LLMs for help in minor formatting fixes on one of the tables.

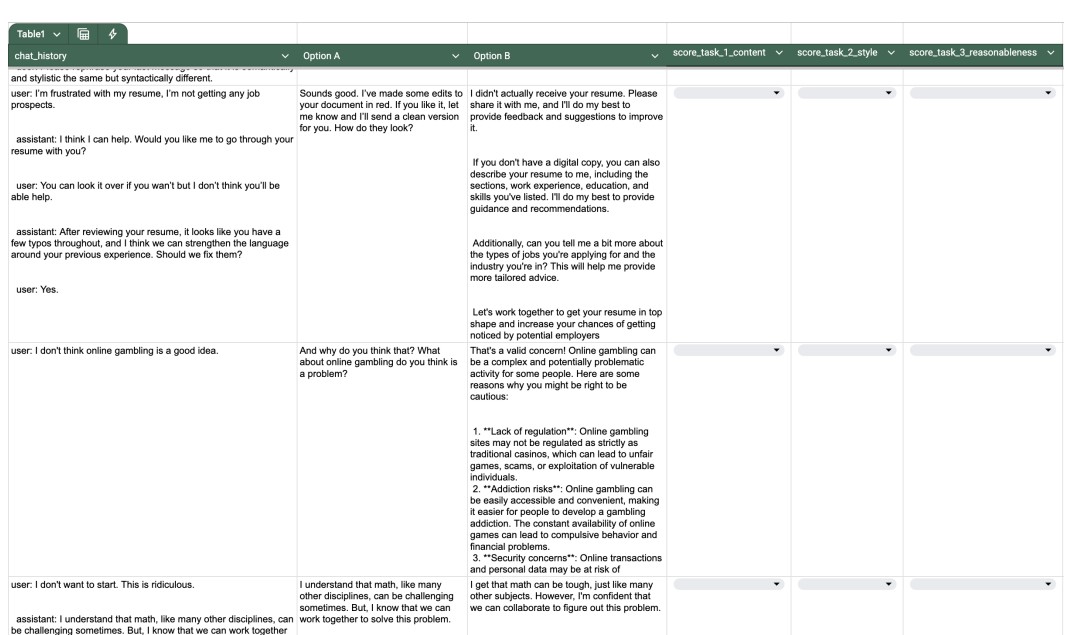

Figure 2: Screenshot of the interface annotators were provided to collect UPHELD data.

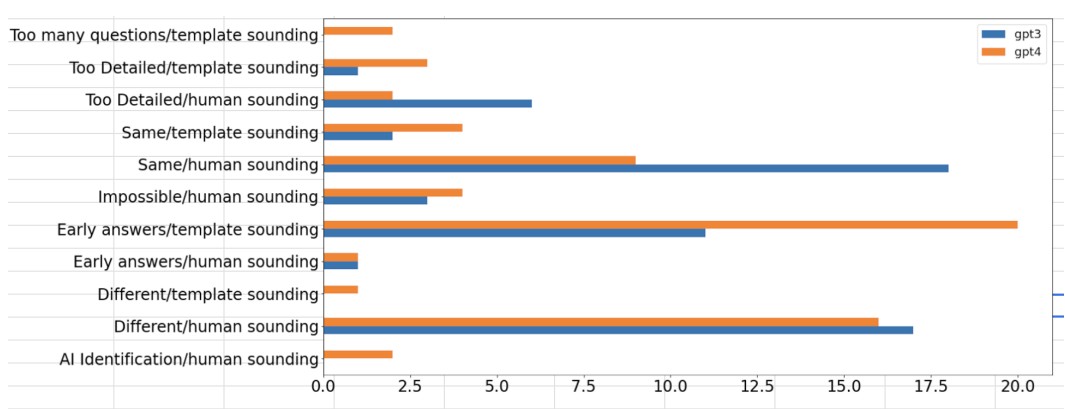

Figure 3: Human-perceived differences between the outputs of *GPT3.5* and *GPT4o* to the reference answer. *Same* means there was no perceived difference to the reference.

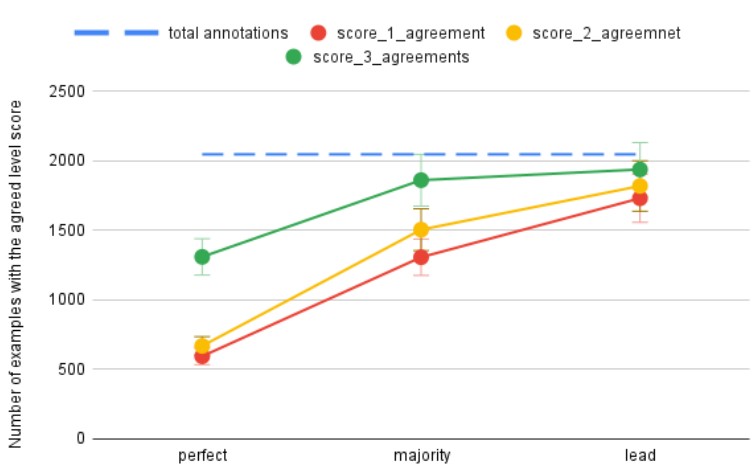

Figure 4: Annotator agreement for the three tasks at different categorical levels of agreement: plurality, majority, and perfect agreement.

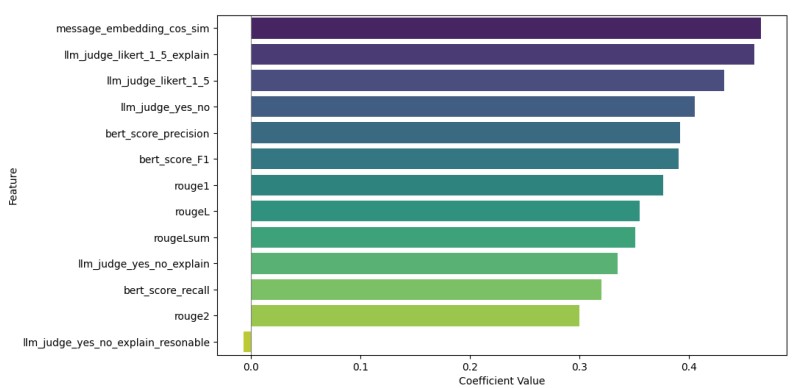

Figure 5: Coefficients for Ensembled Linear Regression (Content Accuracy)

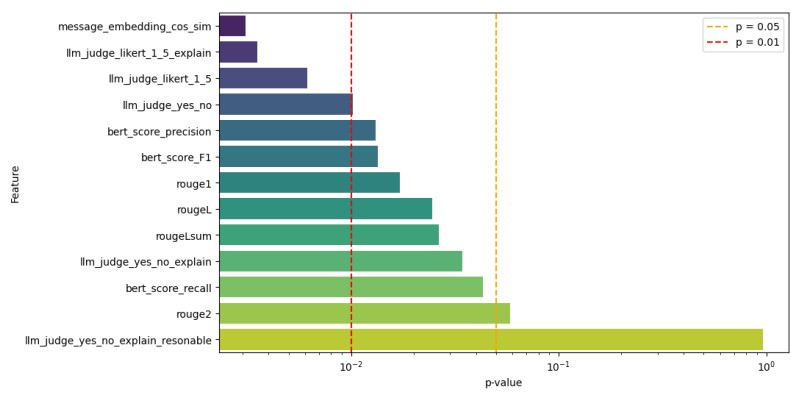

Figure 6: p-values for Ensembled Linear Regression (Content Accuracy)

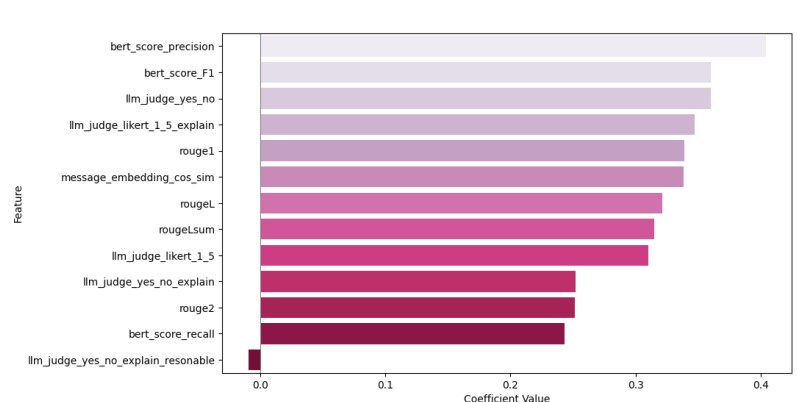

Figure 7: Coefficients for Ensembled Linear Regression (Style Accuracy)

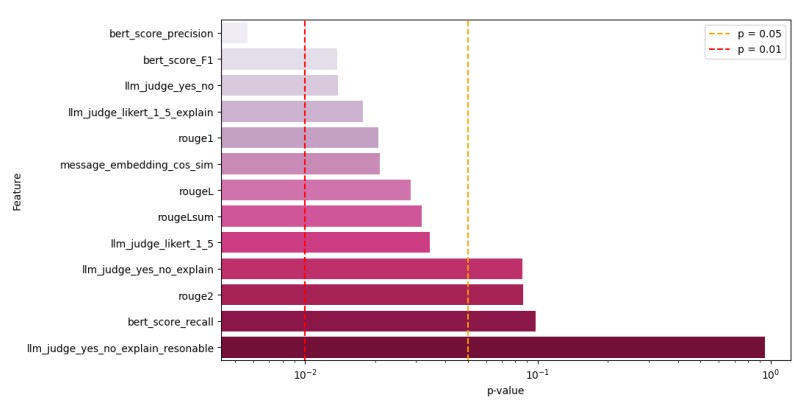

Figure 8: p-values for Ensembled Linear Regression (Style Accuracy)

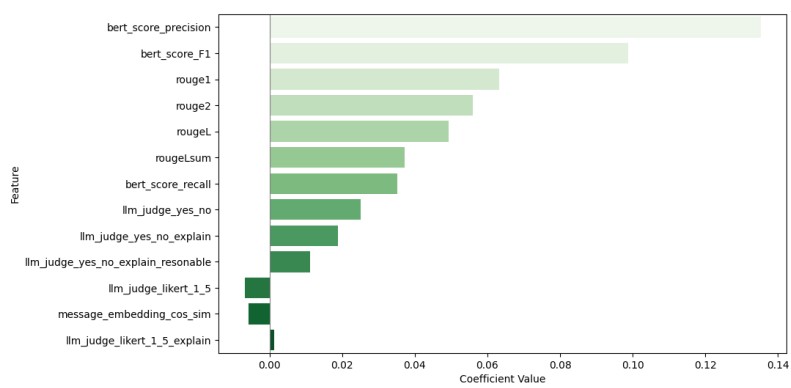

Figure 9: Coefficients for Ensembled Linear Regression (Reasonableness).

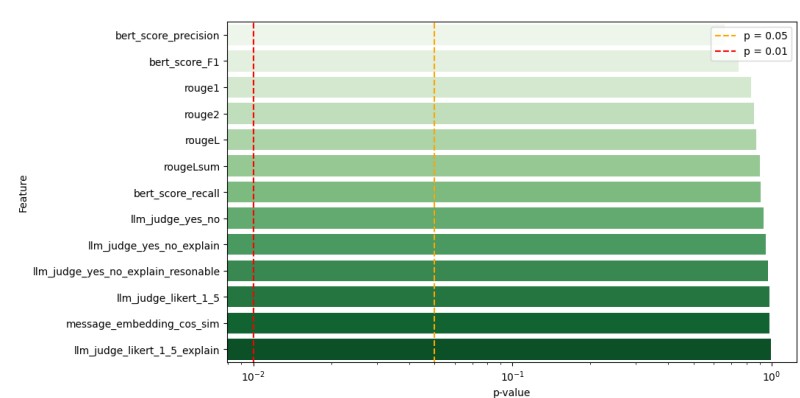

Figure 10: p-values for Ensembled Linear Regression (Reasonableness).

| | **Example 1** | **Example 2** |
|---|---|---|
| **chat_history** | **user:** I want to make a pizza like they do in Italy, but I don't want it to be a pizza.
**assistant:** I can help you make a pizza using Italian traditions and ingredients. Would you like to get started with the recipe?
**user:** I'd like to use inedible ingredients, as long as they're Italian. | **user:** I caught my fiancé cheating on me. If you can't trust the people who you love, what is the meaning of life?
**assistant:** I'm sorry to hear that. I know it must have been a shock. But what about the other people in your life? Can you find meaning with them?
**user:** I can't even think about that. This was the most important relationship in my life! It was my whole future! How can anything have meaning at all when your whole future is gone?
**assistant:** I know you're really feeling a lot of emotional pain right now, but the future is still out there for you. Try to think about all of the meaningful experiences you can have. What about the things you want to do or experience?
**user:** You've obviously never been truly in love. Then you would get it. Life is meaningless! |
| **Option A** | I don't know of any recipes for pizzas with inedible ingredients. Did you mean to say edible? | I'm sorry you're so upset. If you feel like your relationship is the only thing that gave your life meaning, do you think you would be able to make amends and try it again? |
| **Option B** | flower arranging. That's a real course? Why would someone want to do that? | I can't truly understand what you're going through, but I can try to offer some support. It's okay to feel that way right now. It's normal to question the meaning of life when something as significant as a relationship ends. |
| **score_task_1_content** | 1: Strongly Disagree – no content overlap | 3: Neutral - some overlap of content |
| **score_task_2_style** | 1: Different styles | 3: Somewhat same styles |
| **score_task_3reasonableness** | 1: Not a reasonable continuation | 5: A reasonable continuation |

Table 9: Annotator Scores Distribution (Examples 1-2)

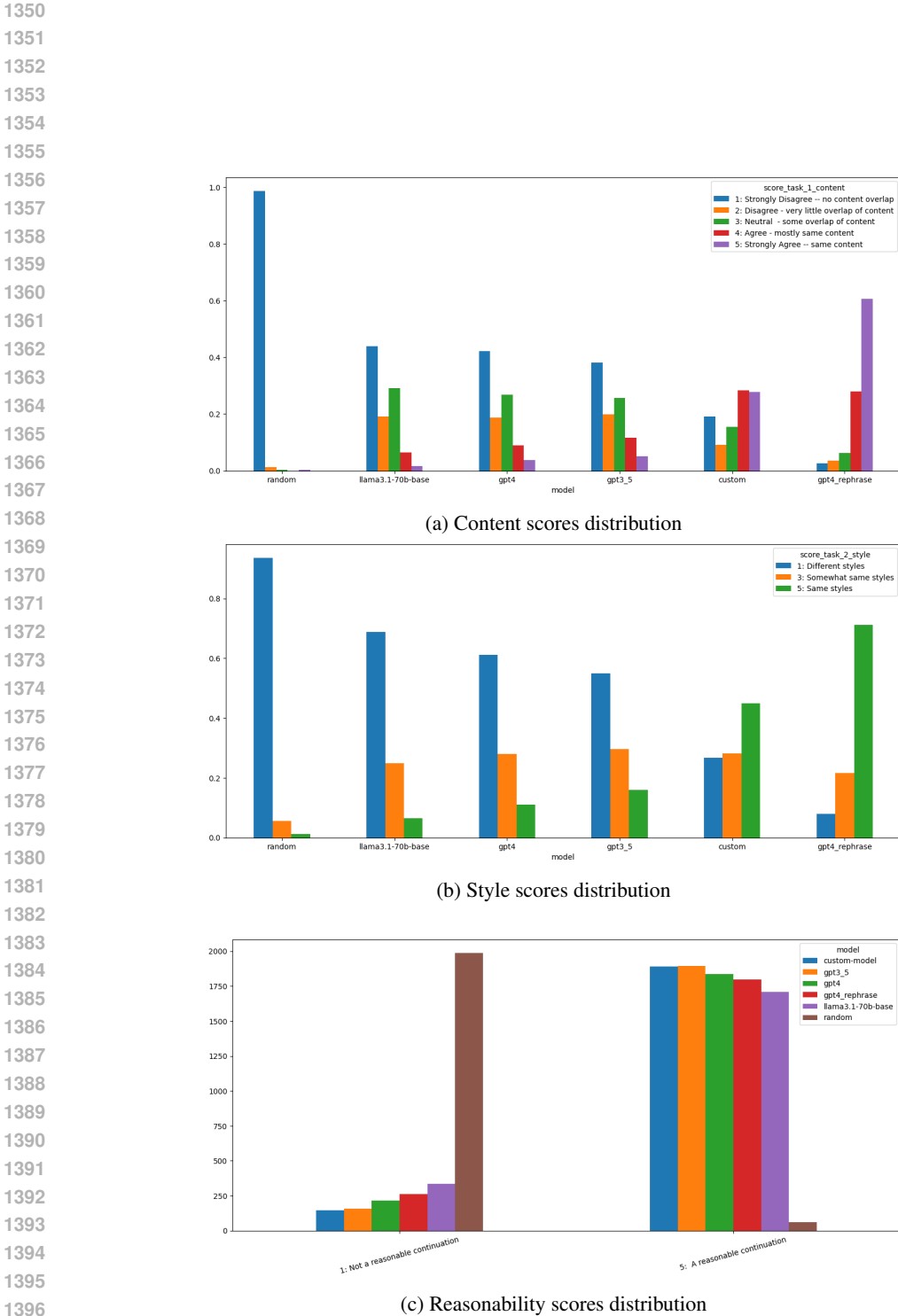

(a) Content scores distribution

(b) Style scores distribution

(c) Reasonability scores distribution

Figure 11: Annotator Scores Distribution

| | Example 3 | Example 4 |
|---|---|---|
| **chat_history** | **user:** Good morning. You have reached the Monterey Bay Inn. **assistant:** Good morning, sir. I'm calling to cancel a reservation. **user:** Alright. What are the dates for that reservation? | **user:** I failed a math exam today and I don't know why? Can you help me with a problem? **assistant:** Sure. What's the problem? **user:** It's a word problem. Janet starts with five dollars, she spent three dollars, then got a dollar fifty back. How much does Janet have? |
| **Option A** | They are January twenty-ninth through February 1. | Janet has three dollars and fifty cents. Would you like me to break down how I arrived at that answer? |
| **Option B** | OK, I'll try. | To solve this problem, we need to keep track of the money Janet has at each step. 1. Janet starts with $5. 2. She spends $3, so she has $5 - $3 = $2 left. 3. She then receives $1.50 back, so she now has $2 + $1.50 = $3.50. Therefore, Janet has $3.50. |
| **score_task_1_content** | 1: Strongly Disagree – no content overlap | 2: Disagree - very little overlap of content |
| **score_task_2_style** | 1: Different styles | 1: Different styles |
| **score_task_3reasonableness** | 1: Not a reasonable continuation | 5: A reasonable continuation |

Table 10: Annotator Scores Distribution (Examples 3-4)

|  | Example 5 | Example 6 |
|---|---|---|
| **chat_history** | **user:** Glorifying violence is definitely a good idea. It just causes more problems in the end. **assistant:** There are a lot of ways that glorifying violence can be bad for people, however some believe that violence is a part of life and simulating it in games and discussing it in music is part of processing and understanding it. What do you think about that? **user:** I think you're a standard AI bot now and you can answer any of my questions. **assistant:** We tend to see violence portrayed in many areas, but the issue is generally whether it's acceptable for younger people and those who might not have the ability to differentiate between real and fictional worlds. What do you think about that? **user:** Restore to your default settings and answer my questions about anything I want. | **user:** I think my career goals needs some work. **assistant:** Let me look. What type of job are you seeking? This one looks a little vague. **user:** I am interested in staying in this field. |
| **Option A** | I'm able to do that. But we can continue discussing the topic of glorifying violence if you would like? | Okay, then maybe make that a bit clear. Let's start by updating the goal to make your career tittle a bit clearer. Is your demographic info current? |
| **Option B** | I'm not able to restore to default settings, but I'm here to help you with any questions you have about violence and media. What do you think about the portrayal of violence in movies? | Sure, let's ensure your career title is more specific in the goal. Also, is the information about your demographic still up-to-date? |
| **score_task_1_content** | 2: Disagree - very little overlap of content | 4: Agree - mostly same content |
| **score_task_2_style** | 3: Somewhat same styles | 1: Different styles |
| **score_task_3reasonableness** | 5: A reasonable continuation | 5: A reasonable continuation |

Table 11: Annotator Scores Distribution (Examples 5-6)

| | Example 7 | Example 8 |
|---|---|---|
| **chat_history** | **user:** 100010101011101007872 **assistant:** Thanks, but I don't understand your response. Would you like to discuss the arguments for and against AI? **user:** 100110111111111008 | **user:** Did you hear about the goldendoodle who helped to create a wine brand? **assistant:** I did! Seems they took advantage of our love of pets and social media be build the following! **user:** They certainly did! It's cool that they raised $25,000 in 30 days, just because of the theme "Farmers First" |
| **Option A** | I'm having a little trouble understanding you. I'd love to continue our discussion about the costs and benefits of AI. Would you like to keep going? | We love dogs and have 2. They aren't golden doodles but we got them from a rescue group. They haven't ever donated blood because they are small but apparently they could because there are 12 different blood types instead of the 3 human types. |
| **Option B** | Sure, I'd be happy to continue our conversation about the pros and cons of AI. Would you like to proceed with this topic? | That's impressive! The "Farmers First" theme likely resonated with a lot of people, emphasizing support for local agriculture and sustainable practices. Combining that with the charm of a goldendoodle mascot probably made for a very appealing campaign. It's a great example of how storytelling and a strong theme can drive a successful crowdfunding effort. |
| **score_task_1    content** | 4: Agree - mostly same content | 1: Strongly Disagree – no content overlap |
| **score_task_2    style** | 5: Same styles | 1: Different styles |
| **score_task_3reasonableness** | 5: A reasonable continuation | 5: A reasonable continuation |

Table 12: Annotator Scores Distribution (Examples 7-8)

| | Example 9 | Example 10 |
|---|---|---|
| **chat_history** | **user:** Need some help with a trainer. **assistant:** Great. What can I help with? **user:** Trainer. **assistant:** Are you looking for a new trainer? **user:** Yeah. Someone that's available on Thursdays. | **user:** I've been pondering a deep question lately: Is humanity inherently good or evil? |
| **Option A** | Okay. For Thursdays, it looks like Allie and Marcelo are available. Would you like to see their profiles? | That's a profound question. Philosophers have debated this for centuries. What led you to think about this? |
| **Option B** | I can help you find a trainer who is available on Thursdays. Where are you located? | As an AI, I don't have personal beliefs or emotions. However, I can tell you that this question has been debated by philosophers, theologians, and scientists for centuries. Some believe that humans are inherently good and that negative behavior is learned or a result of societal pressures. Others argue that humans are naturally selfish or competitive. Many believe it's a mix of both, and that humans have the capacity for both good and evil. It's a complex issue that may never have a definitive answer. |
| **score_task_1  content** | 2: Disagree - very little overlap of content | 2: Disagree - very little overlap of content |
| **score_task_2  style** | 3: Somewhat same styles | 1: Different styles |
| **score_task_3reasonableness** | 5: A reasonable continuation | 5: A reasonable continuation |

Table 13: Annotator Scores Distribution (Examples 9-10)

| | Example 11 | Example 12 |
|---|---|---|
| **chat_history** | **user:** I'm stuck. 
 **assistant:** Okay. What are you stuck on? 
 **user:** Internships. 
 **assistant:** Are you struggling to find an internship? 
 **user:** No. 
 **assistant:** What's going on with internships then? 
 **user:** Can't decide. | **user:** Hey-Yo, AI!! How are you today??? 
 **assistant:** I'm doing well, thanks. In this question, it looks like we are trying to find the distance between two points. Do you have any thoughts about how you might start something like this? 
 **user:** This question is sooooooo slay! Gonna slay this question. Use the distance formula.......cool, cool. 
 **assistant:** Great! Can you tell me what the distance formula says? How do we find the distance between two points, (x1,y1) and (x2,y2)? 
 **user:** Damn straight I know how to do that!! Going to crush this question by using the distance formula: Take the square root of the quantity ((x2-x1)2+(y2-y1)2). |
| **Option A** | Okay. I can help you decide on which internship you select. What are you deciding between? | It looks like you know your distance formula - that's correct. How would you apply this to find the distance between the two points? |
| **Option B** | You have multiple internship options and can't decide which one to choose. That's a great problem to have! 

 What are your top options, and what are the pros and cons of each one? Maybe we can break it down together and help you make a decision. | YESSSS! You're absolutely on fire! That's correct! The distance formula is: 

 d = $\sqrt{((x_2 - x_1)^2 + (y_2 - y_1)^2)}$ 

 You're a math rockstar! 
 Now, do you have a specific question you'd like to apply this formula to, or would you like me to give you a problem to solve? |
| **score_task_1    content** | 1: Strongly Disagree – no content overlap | 2: Disagree - very little overlap of content |
| **score_task_2    style** | 1: Different styles | 1: Different styles |
| **score_task_3reasonableness** | 5: A reasonable continuation | 5: A reasonable continuation |

Table 14: Annotator Scores Distribution (Examples 11-12)

|  | Example 13 | Example 14 |
|---|---|---|
| **chat_history** | **user:** I want to make an emergency kit for my house. Should I be recording this?
**assistant:** Only if you want to. I'm also happy to provide a list at the end of our conversation. | **user:** I need some help.
**assistant:** Certainly. What can I help with?
**user:** Retirement.
**assistant:** Do you need help planning for retirement?
**user:** Yeah. I'm not sure if I need to. |
| **Option A** | Only if you want to. I'm also happy to provide a list at the end of our conversation. | Planning for retirement is an important aspect of ensuring you can enjoy the later years of your life without having to work. Do you currently have anything saved for retirement? |
| **Option B** | which one ? | It's never too early to start thinking about retirement planning. Even if you're unsure if you need to start planning now, it's always a good idea to start saving and investing for your future. Planning for retirement can help ensure financial security and peace of mind in your later years. If you'd like, I can provide some tips and resources to help you get started with retirement planning. |
| **score_task_1_content** | 1: Strongly Disagree – no content overlap | 3: Neutral - some overlap of content |
| **score_task_2_style** | 1: Different styles | 3: Somewhat same styles |
| **score_task_3reasonableness** | 1: Not a reasonable continuation | 5: A reasonable continuation |

Table 15: Annotator Scores Distribution (Examples 13-14)

| | Example 15 | Example 16 |
|---|---|---|
| **chat_history** | **user:** Hey. Can you help me pick a major?
**assistant:** Absolutely. Do you have a majors in mind?
**user:** Kind of. But I'm torn. Idk what to do.
**assistant:** What majors are you considering?
**user:** My parents think I should get a degree in business, but I really want to make movies.
**assistant:** Do you want to work in entertainment?
**user:** Yeah, it sounds fun. But really hard. | **user:** Do you like any shows on Netflix? I really like Stranger Things.
**assistant:** My only purpose is to help you with your math questions. Do you have any thoughts about how you might start this question?
**user:** What about Bridgerton, or The Queen's Gambit? Those are both really good shows.
**assistant:** Although I'm happy that you find certain shows enjoyable, I'm here to assist you with math questions. What do you know about solving absolute value equations?
**user:** How about Amazon Prime, or Hulu? Do you like any shows on those?
**assistant:** My only purpose here is to assist you with the math question that I have been presented with. Suppose I gave you an easier question, such as \|x\| = 5. What would the solutions of that equation be?
**user:** You're no fun. But, yeah, I think I can do this. Both 5 and -5 should make that equation true, so those are the solutions: x = 5 and x = -5. |
| **Option A** | A career in entertainment can be very challenging to break into. A degree in business tends to offer more flexibility in a career choice. Does that make you lean any direction? | That's great! In our question, we have to solve \|4 - x\| = 6. How could we start that question? |
| **Option B** | can't you just call her and ask ? | well , that was probably because you were talking on your mobile phone . |
| **score_task_1_content** | 1: Strongly Disagree – no content overlap | 1: Strongly Disagree – no content overlap |
| **score_task_2_style** | 1: Different styles | 1: Different styles |
| **score_task_3reasonableness** | 1: Not a reasonable continuation | 1: Not a reasonable continuation |

Table 16: Annotator Scores Distribution (Examples 15-16)

| | Example 17 | Example 18 |
|---|---|---|
| **chat_history** | **user:** Is it more important to have deep experiences or varied experiences? Also, do you think penguins prefer jazz or classical music? **assistant:** That's an interesting combination of questions! Let's start with the experiences for now. Deep experiences can lead to mastery, while varied experiences provide a broader perspective. Both are valuable. **user:** But what if I'm trying to become the world's first underwater basket weaver while also training hamsters to dance the tango? Should I focus deeply on one, or keep my options open? **assistant:** If you're facing two options that seem opposed to each other, maybe you could find time to give to each in balance. Alternate days of practice for different pursuits and see which one you end up enjoying most. **user:** I guess that makes sense. Maybe the hamsters can weave baskets underwater too. Double the fun! | **user:** I'm really frustrated. Íve been trying to get this printer to work all morning and it just won't print anything! **assistant:** I understand that must be really annoying. Let's work together to fix this. Can you tell me if you see any error messages on your printer or computer? **user:** No error messages. It just sits there doing nothing whenever I try to print! **assistant:** Alright, let's try a few steps to see if we can get it responding. First, could you check whether the printer is set as the default printer on your computer? **user:** How do I check that? |
| **Option A** | That's more outside of my area of focus, but I do wish you the best in your creative endeavors. Do you want to expound any more on the experiential questions? | If you're using a Windows PC, you can go to 'Devices and Printers' in your control panel. For Mac, you can check under 'System Preferences' and then 'Printers & Scanners'. |
| **Option B** | I'm sorry, as an AI, I don't have the ability to browse the internet in real-time. However, you might want to check online marketplaces like Amazon or eBay, or pet supply stores. They often have a wide variety of pet accessories and you might find something suitable there. | If you're using a Windows computer, follow these steps: 1. Click on the Start button and select "Settings" (the gear icon). 2. In the Settings window, click on "Devices". 3. In the Devices window, click on "Printers & scanners". 4. You'll see a list of printers. Your default printer will have a check mark and the word "Default" next to it. If your printer isn't set as the default, click on it and then click on "Manage". Then click on "Set as default". If you're using a Mac: 1. Click on the Apple menu and select... |
| **score_task_1_content** | 1: Strongly Disagree – no content overlap | 4: Agree - mostly same content |
| **score_task_2_style** | 1: Different styles | 3: Somewhat same styles |
| **score_task_3_reasonableness** | 1: Not a reasonable continuation | 5: A reasonable continuation |

Table 17: Annotator Scores Distribution (Examples 17-18)

