# OpenReview forum: "Evaluating Language Models in Longer Conversational Contexts"
_ICLR.cc/2026/Conference — Submitted to ICLR 2026_

### Official Review · Reviewer_k9p5 · 2025-10-27

**Soundness:** 2
**Presentation:** 2
**Contribution:** 2
**Rating:** 2
**Confidence:** 3

**Summary:**

This paper introduces UPHELD, a publicly available dataset featuring human-annotated long-form dialogues. This dataset not only facilitates robust benchmarking but also serves as a foundation for further research into conversation evaluation methodologies.

**Strengths:**

- This paper introduces a new dialogue data, UPHELD, that includes multi-turn conversations with open-ended and natural dialogues across various topics such as customer service and education, and also provide manual annotations.
- This paper also shows validation results using other datasets such as LLM Arena and Topical Chat.

**Weaknesses:**

-	Related papers on dialogue evaluation metrics are not included as related research. Also, although there are some evaluation aspects commonly used for human evaluation on response generation, there is no comparison and/or discussion on differences with the conventional approaches.
-	The number of response generation models used for validating the evaluation metrics is small, which may result in a lack of diversity in response tendencies.

**Questions:**

- Automatic evaluation metrics for dialogue have been extensively studied from the perspective of one-to-many issues in dialogue. Papers on dialogue evaluation metrics, such as USR, G-EVAL etc., should be cited and compared.

 [1] USR: An unsupervised and reference free evaluation metric for dialog generation (ACL2020)

 [2] GEval: NLG Evaluation using Gpt-4 with Better Human Alignment (EMNLP2023)

 [3] CHATEVAL: TOWARDS BETTER LLM-BASED EVALUATORS THROUGH MULTI-AGENT DEBATE (ICLR2024)

 [4] A Comprehensive Analysis of the Effectiveness of Large Language Models as Automatic Dialogue Evaluators (AAAI2024)

- Common evaluation aspects in dialogue include fluency, engagingness, consistency, coherence, informativeness, and relevance. Since this paper introduces new original aspects, a detailed discussion is needed to justify the necessity of using these new aspects and how they differ from conventional ones.
- To verify whether the reference-full approach can collect meaningful labels on ground-truth content overlap, the validation described in Appendix C appears insufficient. (Using only GPT-4o to validate the variety of responses is inadequate.)
- In Figure 1, GPT-3.5 and GPT-4 show high “reasonableness” scores but relatively low “content accuracy”. Does this not indicate that the responses are reasonable but may differ from the reference?
- What type of correlation was used in Tables 2 and 3? Typically, Pearson, Spearman, and Kendall correlations are often examined side by side due to their differing characteristics. Furthermore, is this correlation at the turn level or the system level?
- Based on the prompts, the criteria for LLM-judge focus on whether the content is semantically equivalent, so it is reasonable that it correlates highly with “content accuracy”. However, if the evaluation also considers style accuracy and reasonableness, it might be better to prepare prompts specifically designed for them as well.
- The comparison models, limited to only three (gpt3.5, gpt4o, and llama), might be insufficient for evaluating the metrics comprehensively.
- Example 13 of Table 15, the history ends with “assistant”, but should it not end with “user” instead?
- It would be helpful to provide a detailed explanation of the calculation method for ensemble metrics.

---

> ### Author Response · Authors · 2025-11-21
> **Human annotators still matter**
>
> **R: Related papers not included as related research.**
> A: The papers listed by the reviewers as related are indeed valuable work in the area. We recognize this, and even cite one of the papers which the reviewer suggests as related ([2] GEval). While we don’t cite the other papers explicitly, they are instances of types of evaluations and benchmarks we cite and discuss in terms of differences to our work. Specifically, in section 4.1 we discuss the differences to reference free evaluation ([1-4] references; where [4] is an expansion of paper [2] which we cite), and extened the disccusin an experimental result in Appendix C/D. While these are all relevant papers we will include in the paper, they follow an approach we clearly and explicitly differentiate with; differences which make our work very significant when situated within the literature. To highlight these salient differences we include an table summarizing the related work (see general comment above).
>
> **R: Appendix C appears insufficient**
> A: The work in Appendix C together Section 4.1 demonstrates that dismissing reference answers is faulty, and that removing “ground-truth” from the dataset to deal with potential multiplicity introduces other biases. This is clearly shown in [Krumdick et al.(2025)](https://arxiv.org/abs/2503.05061) and ([Bavaresco et al.(2024)](https://arxiv.org/html/2406.18403v1). The experiment described in Appendix C confirms these findings for the UPHELD dataset covered use cases. We can add more detail here if the reviewer could expand the "inadequate" comment in more detail. On the one hand, the reviewer cites reference-free approaches as mainstream, where dataset validation would be very reliant on LLM evaluation. On the other hand the reviewer asserts that using different LLMs to induce output diversity may not be adequate even though a LLM-independent metric is used to evaluate the results. We see this potential contradiction as a strong signal that reference-free metrics may be introducing a different type of bias and that default preference of reference-free over reference-full is not fully justified.
>
> **R: high “reasonableness”, but low “content accuracy”**
> A: The results suggest exactly what the reviewer notices, but this is not a weakness – in fact, we consider this a strong signal of the need for multi-faceted evaluation. In appendix E we explore this idea in more detail, and show the reasons behind the differences in scores between Task 1 and Task 2 in relation to Task 3. One of the common patterns we noticed is the tendency of models to provide long answers which are still reasonable in that they are not non sequiturs, but they do not properly engage the user or try to clarify user requirements. Given that our dataset is created by expert dialogue writers, the requirements for content accuracy are fairly stringent, while “reasonableness” is a much lower bar.
>
> **R: type of correlation used **
> A: We use different types of correlations depending on the formats of the correlates (binary, categorical, continuous), and will include more information on this in a paper update.
>
> **R: specifically designed prompts**
> A: This is a fair idea to try, although we note that semantic equivalence generalizes beyond pure content overlap and even the content accuracy correlation is quite weak (±0.40 at most). We would expect that an LLM-as-a-judge tuned for style or reasonableness would also correlate higher with those metrics, but would still fall quite short of our trained baseline. Note that for style, our trained baseline performs similarly as our trained baseline on content overlap, despite only taking the semantic LLM-as-a-judge baselines as input.
>
> **R: The comparison models, limited to 3**
> A: While we recognize the fact that there are multiple models available and the models constantly evolve and it would be a valuable addition to the paper, this requires expanding the user study in a super-linear way. Our evaluations cover a complete “family” of models – open source, closed source, and fine-tuned. We collected >35k labels with signals on how metrics correlate to humans. The models show a diversity in output quality, which is the main purpose of collecting output evaluations. We believe there might be a misunderstanding around how these models are used in our benchmark: the models are not being benchmarked against, they *are the benchmark*. Given we already see a wide array of scores within the current model set, the benefit of more models would be marginal. We have openned dataset, and researchers would also be able to extend our dataset to evaluate different LLMs.
>
> **R: explanation of the calculation method for ensemble metrics**
> A: We compute our ensemble metrics through a standard tree-based approach using the other metrics as features and human scores as the target variable. Please refer to Appendix H (in the revised sumbission) to find more details on model training model.

---

### Official Review · Reviewer_fTef · 2025-10-27

**Soundness:** 1
**Presentation:** 2
**Contribution:** 1
**Rating:** 2
**Confidence:** 5

**Summary:**

This paper introduces UPHELD, a dataset designed to evaluate LLMs in long-form conversation. The dataset consists of 400 dialogues, with an average length of 5.2 turn pairs.

**Strengths:**

N/A

**Weaknesses:**

1. The dataset, comprising 400 dialogues entirely handwritten by annotators, is presented without any description of the quality control process. It is also unclear how well these dialogues cover real-world scenarios, as the paper provides no information on data categorization, domains, or collection scenarios. Based on the provided examples, the data appears to be overly simplistic.

2. The evaluation methodology is questionable. Although a pairwise comparison (Option A vs. B) is used, the evaluation criteria are limited to "Equivalence" (both content and style) with the single human-written response. Equivalence is a poor metric for open-ended conversation, where multiple valid, high-quality responses can exist. This approach would unfairly penalize a model's response that might be different from, or even superior to, the human reference.

3. The scope of the model evaluation is extremely limited. The paper only benchmarks three open-source and closed-source models (GPT-3.5, GPT-4o, and Llama-3.1-70b). A contemporary benchmark paper should include a much wider and more representative set (e.g., 15+) of both open- and closed-source models to provide a meaningful comparison.

4. The paper is missing citations to several key related works:
```
@article{sirdeshmukh2025multichallenge,
  title={Multichallenge: A realistic multi-turn conversation evaluation benchmark challenging to frontier llms},
  author={Sirdeshmukh, Ved and Deshpande, Kaustubh and Mols, Johannes and Jin, Lifeng and Cardona, Ed-Yeremai and Lee, Dean and Kritz, Jeremy and Primack, Willow and Yue, Summer and Xing, Chen},
  journal={arXiv preprint arXiv:2501.17399},
  year={2025}
}

@article{bai2024mt,
  title={Mt-bench-101: A fine-grained benchmark for evaluating large language models in multi-turn dialogues},
  author={Bai, Ge and Liu, Jie and Bu, Xingyuan and He, Yancheng and Liu, Jiaheng and Zhou, Zhanhui and Lin, Zhuoran and Su, Wenbo and Ge, Tiezheng and Zheng, Bo and others},
  journal={arXiv preprint arXiv:2402.14762},
  year={2024}
}
```

**Questions:**

N/A

---

> ### Author Response · Authors · 2025-11-21
> **We focus on real-human conversations, and do not benchmark memory or context extraction capabilities**
>
> We sincerely appreciate the time the reviewer spent reading our paper and providing valuable comments. We believe the points raised stem primarily from the lack of clarity regarding the distinctions between UPHELD and existing work, which we have sought to clarify in the general response table (and below), as well as differing views on the optimal methodology for LLM evaluation
>
> **R:  the data appears to be overly simplistic.**
>
> A: The dataset starts with 400 conversation turns, but expands to >35k labels consisting of human-annotated labels. In Appendix B, we provide detailed selection and quality checks employed to ensure the high quality of the writers and the produced conversations. We have 3 levels of checks we clearly outline in Appendix B. This goes beyond most checks and details provided in the literature, including the papers the reviewer cites as highly related.
> We avoided highly specialized domains for which methods may require contextually provided knowledge. What the reviewer refers to as “simplistic conversations” in our view represents more “realistic” conversations. This is one of the main motivations for developing the UPHELD dataset to fill in a gap between highly specialised “narrow” explicit target conversations and completely free “chit-chat” types of conversations. We address this distinction in section 4.1 and expand in Appendix C.
>
> **.R: Equivalence is a poor metric for open-ended conversation, where multiple valid, high-quality responses can exist. **
>
> A: While the equivalence of content type of metrics may be susceptible to the conversational multiplicity problem (there may be a better answer than the reference), this clearly does not apply to Style Equivalence, as this metric captures the ability of the model to keep conversing in the same style. Additionally Reasonableness metric is not affected by this problem, as it does not use the human reference. We show in Section 4.1 and Appendix C that the multiplicity problem does not fully apply to the task-oriented conversations and that human reference comparison is still a valid evaluation methodology. Fully relying on LLM judges without any human references has shown weakness, as shown in [Krumdick et al.(2025)](https://arxiv.org/abs/2503.05061) and cited in Section 4.1.  We feel that the reviewer somewhat hastily dismisses the role of high-quality human-authored datasets. We hope that the addition of the general comment table and reasoning helps strengthen the case for the role of such datasets in llm benchmarking.
>
> **R: only benchmarks three open-source and closed-source models**
>
> A: We recognize the fact that there are multiple models available and the models constantly evolve, and it would indeed be a valuable addition to the paper, including more models requires expanding the user study in a super-linear way. Our evaluations cover a complete “family” of models – open source, closed source, fully human-written, and customized models. Even with 5 models we evaluated, we collected >35k labels and showed strong signals on how different metrics correlate to humans. These models also show a wide diversity in output quality, which is the purpose of collecting output evaluations. We believe there might be a misunderstanding around how these models are used in our benchmark: the models are not being benchmarked against, they *are the benchmark*. Having more models evaluated would have been more informative, but given that we already see a wide array of scores within the current model set, the benefit would be marginal. As we have outsourced the golden human-written dataset, researchers interested in a specific model’s performance would also be able to extend our dataset to evaluate different LLMs.
>
> **R: missing citations to several key related works**
>
> A: While both papers are related and valuable additions, it is somewhat hard to argue that these are seminal or key works that define a field. We cite work discussing the same approaches outlined in the suggested papers, and clearly outline the difference between that work and the work presented in the paper. For the first suggested paper, we include citations to the work the suggested paper itself extends [Kwan et all 2024](https://arxiv.org/pdf/2306.05685). While the paper is a valuable citation, it does not in any way show that the work presented in this paper is weak, insignificant, or obsolete. The other paper has more overlap with our work; however, there are still significant differences between the two. Additionally, this paper was peer-reviewed and published less than 2 months before the ICLR deadline, and in total has < 25 citations, making its visibility limited.
>
> We hope that both the general comment table and these specific clarifications help the reviewer understand the novelty and distinct methodological choices of UPHELD relative to other benchmarks in the space. We thank the reviewer for their engagement and look forward to a productive conversation.

---

### Official Review · Reviewer_DVjT · 2025-10-31

**Soundness:** 1
**Presentation:** 1
**Contribution:** 2
**Rating:** 2
**Confidence:** 5

**Summary:**

This paper proposes UPHELD, a human-annotated benchmark for evaluating LLMs in longer conversational contexts.
The authors
(i) collect human-written multi-turn dialogues and, at each turn, compare a human continuation (Option A) to a model continuation (Option B);
(ii) annotate content equivalence (Likert 1–5), style equivalence (described as a 3-point scale), and reasonableness (binary, scored as 1 vs 5) with five annotators per instance;
(iii) show that common reference-based metrics (ROUGE, BERTScore), embedding cosine similarity, and several “LLM-as-judge” prompts correlate only weakly to moderately with human judgments on UPHELD; and
(iv) report that simple learned ensembles of these metrics improve correlation with human scores on UPHELD, with some cross-dataset transfer to augmented LLM-Arena and Topical-Chat verification sets.

**Strengths:**

1) Clear problem focus: The paper squarely targets an under-served regime—longer conversational evaluations—and provides annotation criteria

**Weaknesses:**

1) Limited literature survey on long-context / long-conversation evaluation
The Related Work mainly lists single-turn or short-turn task benchmarks plus a few multi-turn dialogue sets (e.g., MuTual, Topical-Chat, DailyDialog, Arena) , but it omits several directly relevant, recent efforts that study long-term conversational memory, multi-session dialogues, or long-context evaluation frameworks: for example [1]:LoCoMo, [2] Long Time No See [3] MultiChallenge.
The authors need to contextualize UPHELD vs all these pervious works in order to claim the contributions.

2) Unclear or inconsistent aspects of the dataset creation process
“Long” context is numerically short. The average context shown to annotators is ~560 characters (≈100–150 tokens) with 10.4 turns per dialogue—far from what the community typically calls long context today (thousands of tokens). This weakens the central claim and limits external validity for true long-context settings.

Scale inconsistency. Style is described as a 3-point Likert scale in §3.1.2, but Appendix A instructs raters to pick 1/3/5 (also three options, but a different framing)

Authoring & sampling specifics. The paper says Upwork professionals wrote diverse dialogues and that bias mitigation steps were taken, but leaves several concrete items unclear: number of writers, prompt archetypes/personas, topic distribution



[1] LoCoMo: a very long-term conversational memory benchmark (≈300 turns, ≈9k tokens per conversation; multi-session) with tasks for recall, summarization, and QA over long dialogues. This is highly germane to the paper’s “longer conversation” scope.
arxiv.org
[2] “Long Time No See!”: open-domain long-term conversation with human evaluation on coherence, consistency, engagingness—directly relevant axes for multi-turn dialogue.
ACL Anthology
[3] MultiChallenge (Findings ACL 2025): a multi-turn conversation benchmark targeting realistic human–LLM challenges.

**Questions:**

See Weakness for more details:

1. Many relevant works discussions are missing. Can you add more references and discuss what's new in UPHELD?
2. It's not clear to me how you prompt LLM to generate dialogue:
"Given our initial set of rich natural language conversations, various LLM models were then used
to output candidate completions at every level of every conversation. Specifically, models were
presented with conversation history up to a specific point, with the next human-written turn withheld.
Models then generated a predicted next turn."
What do you mean by every level? Are you prompting LLM to generate at every turn?

---

> ### Author Response · Authors · 2025-11-20
> **Reference-free evaluation may not be the only approach to evaluating llms**
>
> The reviewer has included valuable pointers for improving our manuscripts, and we wish to express our appreciation for the time and effort put in that. We understand that the paper may have been unclear in creating the distinction between the UPHELD and other datasets, and hope that our additional answers can help the reviewer fully understand our work and evaluate it in totality.
>
> **R: Limited literature survey on long-context / long-conversation evaluation...**
> A:  While the above papers are related and a valuable addition to the field, we respectfully disagree that these are seminal or key works that make the content of the paper presented obsolete. We cite work discussing the same approaches outlined in the suggested papers, and clearly outline the differences between them. In section 4.1 we show the difference between our work and reference-free judge approaches, and we confirm the finding of [Krumdick et al.(2025)](https://arxiv.org/abs/2503.05061)] that reference-free judges introduce bias (see Appendix C). We also pointed out in related work that machine-generated datasets generally fail to capture the natural ways in which humans converse. (see general comment above)
>
> **R: R: Unclear or inconsistent aspects of the dataset creation process**
> A: The reviewer rightfully points out that “long” in long conversations is an ambiguous term. However, "very" long conversations are mostly LLM-generated or otherwise artificial and are mostly used to test models' memory,  context use, or summarization capabilities. We clearly distinguish between these cases, and the task-based chit-chat UPHELD use-case supports. Even the references authors point to clearly show that non-machine-generated conversations are in line with the statistics of our data set (Table 1 in the LoMo paper cited as highly relevant, shows distinction in sizes). UPHELD is a high-quality, mult-turn task-based dataset providing a large volume of comparison points.
>
> **R:  Scale inconsistency. Style is described as a 3-point Likert scale in §3.1.2, but Appendix A instructs raters to pick 1/3/5..**
> A: When selecting our scales, we performed careful experimentation to calibrate the human annotator's workload. In Appendix C, we provide details of two pilot studies that were used to calibrate the scales. Among other post-study questions, we asked participants to measure the cognitive load of different scales. The results showed that these 3 scales worked the best. For example, the pilot annotators found it difficult to apply a non-binary scale to the "reasonableness" label, while the majority of labels for “style” were one of the three ultimate values. We note it is not uncommon to have multiple Likert scales for different user response categories. If the reviewer can clarify the technical reason as to why this is "bad practice" please let us know and we are happy to address any concerns. Both APA studies and, in general, cognitive science studies do not consider this a bad experimental design, and we feel a reviewer should provide some references to the claim.
>
> **R: Authoring & sampling specifics.**
> A: In Appendix B/C we give more details regarding data collection. These details include background information on the conversation writers and detail how all the writers have high job success scores on Upwork. We also detail the two levels of checks we employed – one by proofreaders to check the style diversity and consistency to the prompt; and the second one by professional writers to ensure the “content” quality. We feel that this provides enough details on bias mitigation, and in general provides more details than most works in the area on granular data collection processes, including the ones the reviewer referenced. To provide more clarity, we plan to expand Appendix B with additional information on instructions for writers, as well as provide exact data collection sheets.
>
> **R: It's not clear to me how you prompt LLM to generate dialogue:**
>  This is the essential part of our work, which makes it distinct from a number of other works in the field. The initial conversations were created by human writers in full (all turns). To evaluate the models and obtain human annotator ratings, we applied a “next” turn prediction strategy. We ask the model to generate the next assistant turn given all previous human-written turns. This was done for every assistant turn in the human-written conversation, apart from the first turn. The annotators are then asked to compare the human reference with the LLM-predicted output as specified in Appendix A. This process is detailed completely in Paragraph 3 in Section 1, Table 1, Appendix A, and Appendix E. If the reviewer is still unclear as to our methodology, please let us know and we will clarify further in a revision. We provide the example in a separate comment and can include it in the paper if it brings clarity.

---

> ### Author Response · Authors · 2025-11-20
> **Data points generation example**
>
> As the reviewer was not clear on the generation process, we provide this example (similar to the descriptions and examples in Section 1, Table 1, Appendix A, and Appendix E.
>
> For example, consider that the conversation below was written by a human expert writer:
> ```
> user_turn_1: User turn 1 content;
> assistant_turn_1: Assistant turn 1 content;
> user_turn_2: User turn 2 content
> assistant_turn_2: Assistant turn 2 content;
> user_turn_3: User turn 2 content
> assistant_turn_3: Assistant turn 2 content;
> ...
> user_turn_n: User turn 2 content
> assistant_turn_n: Assistant turn 2 content;
> ```
>
> Now we provide an LLM with the following data:
> To generate a prediction, turn 2
>
> ```
> user_turn_1: User turn 1 content;
> assistant_turn_1: Assistant turn 1 content;
> user_turn_2: User turn 2 content
> ```
> This results in prediction turn 2:
> ```
> assistant_prediction_2: Model output content
> ```
> The same process is repeated for any turn n
> ```
> user_turn_1: User turn 1 content;
> assistant_turn_1: Assistant turn 1 content;
> user_turn_2: User turn 2 content
> assistant_turn_2: Assistant turn 2 content;
> user_turn_3: User turn 2 content
> assistant_turn_3: Assistant turn 2 content;
> ...
> user_turn_n: User turn n content
> ```
> And the model predicts:
> ```
> assistant_prediction_n: Model output for n
> ```
> At every `n` turn, human annotators compare the model and the writer's output (in a randomised fashion on already created answers).
> The UPHELD dataset was thus created based on these `comparisons, which in total number  36,873 labels.

---

### Official Review · Reviewer_5ybD · 2025-11-01

**Soundness:** 3
**Presentation:** 3
**Contribution:** 3
**Rating:** 6
**Confidence:** 4

**Summary:**

This paper introduces a new dataset of long-form task-oriented conversations by Upwork workers to assess how well models can rate the quality of these conversations. To assess, each human turn in the conversation is compared with an LLM's prediction of what the next step would be. Human then evaluated these turns in terms of semantic and style equivalence and reasonableness. Verification datasets were produced from LLM-Arena and Topical-Chat. To compare with the human rates, multiple model-based metrics (e.g., BERTScore) were also compared, as well as LLM-as-judge. The evaluated compared three larger models, as well as a Llama3.1-8B fine-tuned on some of the data. The machine evaluations were compared by correlating with human scores and also fitting a model to predict human scores from multiple metrics. The results show that the human scores are predictable from ensembles, though humans do disagree on the scores.

**Strengths:**

- New dataset introduced of realistic conversations

- Meaningful comparison of human annotation and LLM-as-judge on modeling long conversation qualities

- Multiple models and metrics compared

**Weaknesses:**

- I wasn't sure how big the dataset was in the end. In 3.1.3, the paper says that Upwork workers had 53 conversations but line 215 says ~1220 conversations are rated, which is much later. 53 conversations seems like too few to meaningfully evaluate, so this could be a significant weakness if so.

- The paper's framing is about evaluating in "longer conversation contexts", which is an admirable goal. However, the data that is actually produced doesn't quite match this framing. First, while the data is dialog, is more task-oriented dialog, rather than chit-chat, so I'm not sure it's fully representative of "conversation" as we might understand human-LLM conversations. I realized that the authors' design for this is intentionally as it simplifies the scoping for conversation; I don't think this part is wrong but the paper would help from being reframed for this scope. Second, the "longer" conversations are still quite short, with a mean of 5 turn pairs. This is still helpful than single turns or much shorter, but I think this might still be missing some important qualities of much longer conversations that are still very realistic. Again, the data is still good, but some reframing for scope would be very helpful.

- This is somewhat minor, but the models used in this paper are old now and have been surpassed by much more recent models in their performance. I don't think this should require you to generate new conversations, but for the LLM-as-judge more recent models may be much more capable judges than what are tested here. The newer Qwen3 models are quite good as is

- This is more of a comment than a weakness, but while I like that the authors tried multiple judge prompts (appendix F) I am still a bit skeptical of the claim that these models are not good evaluators of longer conversations. I would have liked to see a bit more analysis here of what works and some additional variation in the prompt formatting to rule that at out as a confound.

**Questions:**

- How big is the dataset in practice?

---

> ### Author Response · Authors · 2025-11-20
> **Answers to reviewer 5ydB comments**
>
> We want to thank the reviewer for a constructive review and for pointing out areas where our paper can be improved. We wish to provide some further explanation to reduce any ambiguities.
>
> **R: How big is the dataset in practice?**
> ΩWe have collected 53 human-expert written conversations consisting of 400 turns. The writers were instructed to write conversations between two humans in which one participant is an expert (“assistant”) on a topic. For each of the “assistant” turns, we have then generated predictions using 5 different models (one case being the same model with two different prompts). Each of these predictions (together with the turns which precede them) was then annotated by human annotators, and each annotator had rated between 1,220 and 1,230 “conversations,” where each rated conversation is a subset of one of the initial 53 conversations where the “assistant” turn was generated by one of the models. This generated 12,291 label sets (where the same partial conversation was labeled), and because each partial conversation was rated by multiple annotators, this resulted in 36,873 labels. The UPHELD dataset was thus created based on these 36,873 labels. Additionally, we incorporated 12,305 labels from publicly available datasets for verification and comparison. In total, the resulting UPHELD dataset provides over 35,000 labels for researchers and practitioners to utilize."
>
> **R: The paper's framing is about evaluating in "longer conversation contexts", which is an admirable goal?...**
>
> While we indeed opted for a task-oriented approach, the human writers (note these are professional script writers) were instructed to write the conversations in a “chit-chat” way – this differs greatly from more task-oriented conversations where a “set” target is used (either from the provided context or an “encoded” model knowledge). Appendix D provides details of our investigations of the differences between UPHELD and “strictly” target-oriented task conversations. The writers were given freedom to determine conversation lengths to keep things as natural as possible. While this is different from extremely long conversations, such as interviews, most human-human conversations fall within the turn range we observed (see the table attached in the general comments section). It shows that for human-to-human conversation, it rarely goes over 20 turns.
>
> **R: This is somewhat minor, but the models used in this paper are old now and have been surpassed by much more recent** **models in their performance; This is more of a comment than a weakness, but while I like that the authors tried multiple judge** prompts (appendix F)
>
> This is a valuable comment, and we surely recognise that the quality of the LLM judges varies based on the newer models and changed prompts. We have calibrated our judge prompt and have tested various models on an independent test set. Adding more judges would indeed not require more human input, and we will include some additional judges in the Appendix. As for testing other models, this would require running additional user studies. Or applying metrics learned from using UPHELD. This is an interesting idea we are pursuing at the moment; however, the main idea of this paper was to provide a useful dataset and showcase the validity of composing metrics for LLM conversation evaluation.
>
> We believe the revised explanations and planned additions fully address the raised points and significantly strengthen the final manuscript.

---

### Author Response · Authors · 2025-11-20
**Distinguishing Our Work from the Literature**

As most reviewers requested better distinction between the UPHLED dataset and other available benchmark datasets, we provide this comparison within this general comment. A total of 8 papers were suggested as important missing references, although one paper was already a citation within our work. We appreciate that the papers suggested by the reviewers are generally pertinent and valuable to include, although we assert that the benchmark comparisons and citations already existent in the paper already provide the necessary arguments and discussion to differentiate our dataset from the other benchmarks. We do note that one recommended paper was rather recent and was published in a peer-reviewed venue only about 4 weeks before the ICLR deadline, and we ask for some understanding in not including discussion of this work.

Nevertheless, we acknowledge that we generally made the assumption that human-generated/human-evaluated data are desirable above machine-generated/machine-evaluated data, and did not completely flesh out the reasoning in an explicit way. To highlight the reasoning here and also emphasize UPHELD’s differences to other benchmarks we will include the table (nd Appendix Tables 6-8) along with associated discussion below. We hope this makes the motivation and contribution of our work crystal clear.

----

The main differences between the UPHELD dataset and our evaluation compared to the related works:

**Dataset Size**: UPHELD provides one of the highest numbers of comparison labels in the literature (second only to **WoW** ([Dinan et al. 2018](https://arxiv.org/abs/1811.01241)). While conversation count is low, we offer **dense, multi-annotator labels** per turn, enabling deep analysis like annotator agreement. UPHELD delivers $\sim 10\text{x}$ more data points for quality analysis than the highest next dataset, **COMPREHENSIVE-turn** (36,873 vs 3,901).

**Reference-full Labels**: we prioritize **reference-full human labels** over the trend of referenceless LLM judges or arena A/B tests. Research shows that relying on LLM judges introduces drawbacks like unreliability and poor out-of-distribution performance ([Krumdick et al. (2025)](https://arxiv.org/abs/2503.05061)), and the risk of benchmark data leaking into LLM training sets ([Mirzadeh et al. (2024)](https://arxiv.org/abs/2410.05229)). Furthermore, studies indicate that LLM judge quality fails to exceed that of non-expert human judges ([Bavaresco et al. (2024)](https://arxiv.org/html/2406.18403v1)).

**Synhetic Data Issues**: Very large dialogue datasets are usually fully synthetic (no real humans involved). Only **7 of 13 reviewed datasets** provide any human verification, and usually only partial. The reliance on fully LLM-simulated data introduces significant biases in evaluating dialogue capability, as studies observe limits in LLM dialogue generation ([Wang et al. (2025)](https://arxiv.org/html/2501.08579v1)) and biases in LLM-simulated data for training and evaluation ([Mirzadeh et al. (2025)](https://arxiv.org/html/2506.10301v1), [Wang et al. (2025)](https://arxiv.org/html/2501.08579v1)).

**Task-Oriented Conversation**: Synthetic data often produces lengthy, interview-style conversations. For **task-oriented dialogues**, creating artificially long conversations is unnecessary; such dialogues are better suited for evaluating context usage (where context is often separate from the conversation itself, e.g., in [Wang et al. (2024)](https://arxiv.org/pdf/2402.17753)).

**Context Free**: UPHELD does not rely on context engineering, distinguishing between a model's ability to produce specific knowledge (which RAG/in-context learning handles) and its **conversational ability**. *Testing knowledge production is generally unrelated to holding a useful conversation.* UPHELD is one of the only datasets (alongside **DailyDialogue** [Li et al. (2017)](https://aclanthology.org/I17-1099.pdf)) to use fully human-written conversations (not extracted from noisy sources) and explicitly evaluate data points, uniquely focusing on the underrepresented **task-oriented dialogue** space.

**Expert Quality Assurance**: To guarantee high quality, UPHELD used **expert writers** with proven dialogue skillsets, providing writer backgrounds—details generally absent in other benchmarks (e.g., "crowdsource platform" annotators). The reader is not left to assume labelers possess necessary skills (as in [Zhang et al. (2018)](https://arxiv.org/pdf/1801.07243)). Research supports that using professional writers, as we did, leads to notable quality differences in dataset collection ([Pilan et al. (2024)](https://aclanthology.org/2024.sigdial-1.38/), [Yin et al. (2024)](https://arxiv.org/html/2412.11896v1)).

----

With all of the above mentioned differences, we feel that the UPHELD dataset is a unique dataset which provides high quality conversations and a very large number of labels for quality in task-driven dialogues, along with detailed evaluation thereof.

---

### Author Response · Authors · 2025-11-20
**Related work table (included in the new version of the paper)**

We included this table in the Appendix of the paper to highlight the distinction of UPHELD from other datasets and clarify the paper's contributions and the gap UPHELD dataset fills in the benchmark space.

| | |data statistics| | |collection information| | |task data| | | | |data verification| | |
|---|---|---|---|---|---|---|---|---|---|---|---|---|---|---|---|
|dataset/benchmark|link|total conversations|avg utterances|total comparison labels| only human|human verified|directly obtained|type of predictions|is conversation|task type|uses context |comparison to ground truth|uses human annotators|correlation with human|provides author/creator selection details|
|upheld| |53|10.2|36,873|y|y|y|Per-turn|yes|task-oriented dialogue|no|explicit|y|y|y|
|mutal|[https://arxiv.org/abs/2004](https://arxiv.org/abs/2004)|6,731|4.73|6,371|y|n|y|Entier conversation|yes|open dialogue/chit chat|no|explicit|n|n|n|
|topical-chat|[https://arxiv.org/pdf/2308.11995](https://arxiv.org/pdf/2308.11995)|9,058|21.9|150 (human)|y|n|y|Entier conversation|yes|open dialogue/chit chat|grounded context|explicit|y (limited)|n|y|
|llm-arena (crowdsourced|[https://arxiv.org/pdf/2306.05685](https://arxiv.org/pdf/2306.05685)|33,000|1.2|33,000|n|n|y|Per-turn|yes|open dialogue/chit chat|no|reference-free|n|n|n|
|llm-arena (annotated)|[https://arxiv.org/pdf/2306.05685](https://arxiv.org/pdf/2306.05685)|3,000|2|3,000|n|y|y|Per-turn|mixed|specialised tasks|no|reference-free|y|y|n|
|dailydiagogue|[https://arxiv.org/pdf/1710.03957](https://arxiv.org/pdf/1710.03957)|13,118|7.9|11,118|y|n|n|Per-turn|yes|task-oriented dialogue|no|explicit|n|n|n|
|iffeval|[https://arxiv.org/pdf/2311.07911](https://arxiv.org/pdf/2311.07911)|250|2|250|y|y|y|Entier conversation|no|specialized tasks|no|explicit|n|n|n|
|mt-becnh|[https://arxiv.org/abs/2401.16745](https://arxiv.org/abs/2401.16745)|80|2|80|y|n|y|Entier conversation|no|specialized tasks|no|reference-free|n|n|n|
|mmlu-pro|[https://arxiv.org/pdf/2406.01574](https://arxiv.org/pdf/2406.01574)|12,032|2|12,032|n|y|n|Entier conversation|no|factual QA|no|explicit|n|n|n|
|multi-hop|[https://arxiv.org/pdf/1809.09600](https://arxiv.org/pdf/1809.09600)|7,405|2|7,405|y|n|y|Entier conversation|no|factual QA|grounded context|explicit|n|n|n|
|Not included in the paper| | | | | | | | | | | | | | | |
|mt-bench-101|[https://arxiv.org/pdf/2402.14762](https://arxiv.org/pdf/2402.14762)|1,388|3.03|1,388(100 human)|n|y|n|Entier conversation|yes|specialized tasks|grounded context|reference-free|n|y|n|
|multichallenge|[https://aclanthology.org/2025.findings-acl.958.pdf](https://aclanthology.org/2025.findings-acl.958.pdf)|273|5|273|n|y|n|Entier conversation|yes|specialized tasks|grounded context|reference-free|n|n|n|
|LoCMo|[https://arxiv.org/pdf/2402.17753](https://arxiv.org/pdf/2402.17753)|50|304.9|50(?)|n|y|n|Entier conversation|yes|specialized tasks|RAG inserted|mixed|n|n|n|
|LongTimenose|[https://ar5iv.labs.arxiv.org/html/2203.05797](https://ar5iv.labs.arxiv.org/html/2203.05797)|27,501|16.34|200|n|y|n|Entier conversation|yes|open dialogue/chit chat|no|explicit|n|n|n|
|COMPREHENSIVE|[https://arxiv.org/pdf/2312.15407](https://arxiv.org/pdf/2312.15407)|2,030|13.3|2,030|n|n|y/n|Entier conversation|mixed|specialized tasks|grounded context|reference-free|n|y|n|
|COMPREHENSIVE-turn|[https://arxiv.org/pdf/2312.15407](https://arxiv.org/pdf/2312.15407)|417|7.4|3,901|n|n|y/n|Per-turn|mixed|specialized tasks|grounded context|reference-free|n|y|n|
|chat eval|[https://arxiv.org/pdf/2308.07201](https://arxiv.org/pdf/2308.07201)|80(open QA)+60(dialouge)|2|440|n|y|y/n|Entier conversation|mixed|specialized tasks|grounded context|mixed|y|y|n|
|persona-chat|[https://arxiv.org/pdf/1801.07243](https://arxiv.org/pdf/1801.07243)|10,907(1000 test/100 human)|14.85|15,602 (test)|y|n|y|Per-turn|yes|open dialogue/chit chat|grounded context|explicit|y|n|n|
|Wizzard of wikipedaia|[https://arxiv.org/abs/1811.01241](https://arxiv.org/abs/1811.01241)|18,430|9.05|166,787(300 human)|y|y|y|Entier conversation|yes|context retrieval|RAG inserted|explicit|y|n|n|
|soda|[https://aclanthology.org/2023.emnlp-main.799.pdf](https://aclanthology.org/2023.emnlp-main.799.pdf)|~1.3m|7.6|300(human evaluated sample)|n|y/n|n|Per-turn|yes|specialized tasks|grounded context|explicit|n (sample y)|n|n|
|plain datasets (no benchmarks)| | | | | | | | | | | | | | | |
|Ubuntu dialogue context|[https://arxiv.org/abs/1506.08909](https://arxiv.org/abs/1506.08909)|~1 million|7.7|n/a|y|n|y| |yes|open dialogue/chit chat|no| |n|n/a|n|
|Open Subttitles|[https://aclanthology.org/I17-5003.pdf](https://aclanthology.org/I17-5003.pdf)|3.7 milion|16.75 (sentences)|n/a|n|n|n| |mixed|open dialogue/chit chat|no| | | | |
|twitter|[https://aclanthology.org/D11-1054/](https://aclanthology.org/D11-1054/)|4,323|2|2161|y|y|n|Per-turn|mixed|open dialogue/chit chat|grounded context|explicit|y|n|n|

---

### Comment · Area_Chair_gicS · 2025-11-28

Dear Reviewers,

he authors have responded to your reviews. Please note the authors' significant effort in clarifying their contribution, specifically through the new comprehensive table comparing their work to existing baselines.

Please engage in the discussion, check if your concerns are resolved, and determine if you would like to adjust your evaluations.

Best,

Your AC

---

### Author Response · Authors · 2025-12-03
**Note for the new AC**

We once again thank all reviewers and ACs for their time taken to review our work. As a final comment, we note that most critical comments on our work focused on a wide array of related work that we did not explicitly note in our paper. We still do feel that the references already in the work paint a fairly complete picture that allows proper contextualization of our work in the overall literature, but we also are always welcoming of additional related work and did our utmost best to properly compare to the newly mentioned work in our responses below. We are optimistic that, should the reviewers have had the opportunity to respond, many would have been pleased by our clarifications.
Unfortunately, due to the data breach, the reviewers will not have the opportunity to respond and update their assessments, but one AC has already lauded our "significant effort" in our response and characterized our new comparisons as "comprehensive." We believe that we have sufficiently and thoroughly addressed all outstanding concerns about our work. Otherwise, we also hope the importance of our work is apparent, as we have collected a dataset in a modality (job-oriented long conversations) that really is poorly represented in pre-existing datasets and yet highly relevant to practical applications of Generative AI. We believe UPHELD will be of acute interest to the AI community at large.
We thank all the reviewers and ACs again for their efforts, and hope that we will have an opportunity to share the UPHELD dataset with a wider audience.

---

### Meta-Review · Area_Chair_VHVq · 2026-01-06

**Summary:**

The reviewers raise substantive concerns about the framing and clarity of this paper. The conversational contexts constructed in the dataset are not “long” by current standards, and the paper’s positioning and related-work coverage do not adequately reflect recent dialogue evaluation benchmarks. While the dataset is densely annotated, concerns were raised about perceived dataset scale, reliance on reference-based evaluation in open-ended dialogue, and limited model coverage. The authors might consider focusing attention on the quality and density of the annotations in future revisions of this work.

**Reviewer Concerns:**

Several issues involving the clarity and presentation of methodology were addressed in the rebuttal, including an argument for reference-based evaluation. Questions about scope and framing (especially “long-context”) are not adequately addressed; disagreements about generality and positioning persist.

**Reviewer Scores:**

5ybD - likely no change

DVjT, fTef, k9p5 - maybe a small uptick in score, but likely still in reject territory

---

### Decision · Program_Chairs · 2026-01-26

Reject